# Testing and controlling for horizontal pleiotropy with probabilistic Mendelian randomization in transcriptome-wide association studies

Zhongshang Yuan[1,2], Huanhuan Zhu[2], Ping Zeng [3], Sheng Yang[2], Shiquan Sun [2], Can Yang[4], Jin Liu [5] & Xiang Zhou [2,6✉]

Integrating results from genome-wide association studies (GWASs) and gene expression studies through transcriptome-wide association study (TWAS) has the potential to shed light on the causal molecular mechanisms underlying disease etiology. Here, we present a probabilistic Mendelian randomization (MR) method, PMR-Egger, for TWAS applications. PMR-Egger relies on a MR likelihood framework that unifies many existing TWAS and MR methods, accommodates multiple correlated instruments, tests the causal effect of gene on trait in the presence of horizontal pleiotropy, and is scalable to hundreds of thousands of individuals. In simulations, PMR-Egger provides calibrated type I error control for causal effect testing in the presence of horizontal pleiotropic effects, is reasonably robust under various types of model misspecifications, is more powerful than existing TWAS/MR approaches, and can directly test for horizontal pleiotropy. We illustrate the benefits of PMR-Egger in applications to 39 diseases and complex traits obtained from three GWASs including the UK Biobank.

[1] Department of Biostatistics, School of Public Health, Cheeloo College of Medicine, Shandong University, 250012 Jinan, Shandong, China. [2] Department of Biostatistics, University of Michigan, Ann Arbor, MI 48109, USA. [3] Department of Epidemiology and Biostatistics, Xuzhou Medical University, 221004 Xuzhou, Jiangsu, China. [4] Department of Mathematics, Hong Kong University of Science and Technology, Hong Kong, China. [5] Centre for Quantitative Medicine, Program in Health Services and Systems Research, Duke-NUS Medical School, Singapore 169857, Singapore. [6] Center for Statistical Genetics, University of Michigan, Ann Arbor, MI 48109, USA. ✉email: xzhousph@umich.edu

Genome-wide association studies (GWASs) have identified many SNPs associated with common diseases and disease related traits. Parallel expression quantitative trait loci (eQTL) mapping studies have also identified many *cis*-acting SNPs associated with gene expression level. Integrating the existing association results from both GWASs and eQTL mapping studies has the potential to shed light on the molecular mechanisms underlying disease etiology. Several statistical methods have been recently proposed for such integrative analysis. For example, PrediXcan[1] performs a weighted SNP set test in GWAS using SNP weights inferred from eQTL studies. TWAS[2] infers the association between gene expression and disease trait by leveraging *cis*-SNP information. SMR[3] or GSMR[4] directly tests the causal association between gene expression and disease trait under a Mendelian randomization (MR) framework, using either a single instrument or multiple independent instruments. While each of these integrative methods was originally proposed to solve a different problem, as we will show here, all of them can be viewed as a two-sample MR method with different modeling assumptions. Because of their relationship to MR, these methods effectively attempt to identify genes causally associated with diseases or complex traits in the context of transcriptome-wide association studies (TWAS).

MR analysis is a form of instrumental variable analysis for causal inference[5]. MR aims to determine the causal relationship between an exposure variable (e.g., gene expression) and an outcome variable (e.g., complex trait) in observational studies. MR treats SNPs as instrumental variables for the exposure variable and uses these SNP instruments to estimate and test the causal effect of the exposure on the outcome. MR methods have been widely applied to investigate the causal relationship among various complex traits[6], and, through a two-sample design, can be easily adapted to settings where the exposure and outcome are measured on two different sets of individuals. However, MR analysis for TWAS applications is not straightforward and requires the development of new methods that can accommodate two important features of TWAS analysis.

First, both GWASs and eQTL mapping studies collect SNPs that are in high linkage disequilibrium (LD) with each other. Traditional MR methods, such as the random effects version or the fixed effect version of the inverse variance weighted regression[7], MR-Egger[8], median-based regression[9], SMR[3], or GSMR[4], can only make use of a single SNP instrument or multiple independent SNP instruments. Handling only independent SNPs is restrictive, as most exposure variables/molecular traits are polygenic/omnigenic in nature and are influenced by multiple SNPs that are in potential LD with each other. Consequently, incorporating multiple correlated SNPs can often help explain a greater proportion of variance in the exposure than using independent SNPs and improve MR power[5,10–12]. Due to the benefits of using multiple correlated instruments, most TWAS methods (e.g., PrediXcan[1], TWAS[2], CoMM[13], DPR[14], TIGAR[15]) rely on polygenic modeling assumptions to incorporate all *cis*-SNPs that are in high LD for TWAS applications. By incorporating all *cis*-SNPs, as we will show below, these methods can lead to substantial power improvement over standard MR approaches that use only a few independent SNPs. Unfortunately, many TWAS methods rely on a two-stage MR inference procedure: they estimate SNP effect sizes in the exposure study and plug in these estimates to the outcome study for causal effect inference. The two-stage inference procedure in MR fails to account for the uncertainty in parameter estimates in the exposure study, which can lead to biased causal effect estimates and power loss[5,11,13]. Therefore, it is important to incorporate multiple correlated instruments in a likelihood inference framework for MR analysis in TWAS applications.

Second, perhaps more importantly, SNP instruments exhibit pervasive horizontal pleiotropic effects[16]. Horizontal pleiotropy occurs when a genetic variant affects the outcome variable through pathways other than or in addition to the exposure variable[17]. Horizontal pleiotropy is widely distributed across the genome, affects a wide spectrum of complex traits, and can be driven by LD and extreme polygenicity of traits[16,18]. Despite its wide prevalence, however, only a limited number of MR methods have been developed to test and control for horizontal pleiotropy; even fewer are applicable for TWAS applications. For example, some existing methods (e.g., MR-PRESSO[16]) test for horizontal pleiotropic effects without directly controlling for them. Some methods (e.g., CaMMEL[19]) control for horizontal pleiotropic effects without directly testing them[20,21]. Some methods (e.g., Egger regression[8,22], GLIDE[23], GSMR[4], MR-median method[9], profile score approach[24], MRMix[25], and Bayesian MR[26,27]) test and control for horizontal pleiotropic effects, but can only accommodate independent instruments. As far as we are aware, there is only one two-sample MR method currently developed for testing and controlling for pleiotropic effects in the presence of correlated instruments: LDA MR-Egger[28]. Unfortunately, as we will show below, LDA MR-Egger cannot handle realistic LD pattern among *cis*-SNPs for TWAS applications.

Here, we develop a generative two-sample MR method in a likelihood framework, which we refer to as the probabilistic two-sample Mendelian randomization (PMR), to perform MR analysis using multiple correlated instruments for TWAS applications. Within the PMR framework, we focus on a particular horizontal pleiotropy effect modeling assumption based on the burden test assumption commonly used for rare variant test. This particular horizontal pleiotropy effect effectively generalizes the Egger-regression assumption commonly used for MR analysis to correlated instruments. We refer to our method as PMR-Egger. With simulations and real data applications, we show that PMR-Egger provides calibrated type I error for causal effect testing in the presence of horizontal pleiotropic effects, is more powerful than existing MR approaches, can directly test for horizontal pleiotropy, and is scalable to hundreds of thousands of individuals.

## Results

**Method overview**. PMR-Egger is described in the Methods, with technical details provided in the Supplementary Notes. PMR-Egger relies on a MR likelihood framework (Supplementary Fig. 1) that unifies many existing TWAS and MR methods (Table 1), facilitating the understanding of these existing TWAS/MR approaches. For TWAS applications, PMR-Egger examines one gene at a time and estimates and tests its causal effect on a trait of interest. PMR-Egger models multiple correlated instruments, performs MR inference in a maximum likelihood inference framework, is capable of testing and controlling for horizontal pleiotropic effects commonly encountered in TWAS, and is computationally efficient (Table 2).

**Simulations: testing and estimating the causal effect**. We performed simulations to examine the effectiveness of PMR-Egger, and compared it with existing MR approaches. Simulation details are provided in the Methods. Our first set of simulations is focused on causal effect testing. We compared PMR-Egger with five methods that include SMR, PrediXcan, TWAS, CoMM, and LDA MR-Egger. We first examined type I error control of different methods under the null ($\alpha = 0$). In the absence of horizontal pleiotropic effects, PMR-Egger, together with PrediXcan, TWAS, and CoMM, all provides calibrated type I error (Fig. 1a). In contrast, SMR produces overly conservative/deflated *p*-values

**Table 1 Summary of some existing MR methods.**

| | Design | Instrumental variable | β-effect assumption | γ-effect assumption | Estimation procedure |
|---|---|---|---|---|---|
| PrediXcan[1] | Two-sample | Correlated | Elastic net | N/A | Two-stage |
| TWAS[2] | Two-sample | Correlated | BSLMM | N/A | Two-stage |
| SMR[3] | Two-sample | Univariate | Fixed effect | N/A | Two-stage |
| GSMR[4] | Two-sample | Independent | Fixed effect | N/A | Two-stage |
| MR-Egger[8] | Two-sample | Independent | Fixed effect | Equal effect size | Two-stage |
| CoMM[13] | Two-sample | Correlated | Normal | N/A | MLE |
| CaMMEL[19] | Two-sample | Correlated | Fixed effect | Normal | Variational Bayes |
| Kang et al.[20] | One-sample | Correlated | Fixed effect | Lasso | Two-stage |
| MRMix[25] | Two-sample | Independent | Normal mixture | Normal mixture | Estimating equation |
| Berzuini et al.[26] | One-sample | Correlated | Fixed effect | Horseshoe | MCMC |
| LDA MR-Egger[28] | Two-sample | Correlated | Fixed effect | Equal effect size | Two-stage |
| DPR[14] | Two-sample | Correlated | Latent Dirichlet process | N/A | Two-stage |
| TIGAR[15] | Two-sample | Correlated | Latent Dirichlet process | N/A | Two-stage |
| PMR-Egger | Two-sample | Correlated | Normal | Equal effect size | MLE |

Methods are categorized based on the experimental design (two-sample vs one-sample), the characterizes of selected instrumental variables (univariate vs multiple independent vs multiple correlated), β-effect size assumption, γ-effect size assumption, estimation/inference procedure (ratio-based vs two-stage estimation vs maximum likelihood vs Bayesian), and input data type (individual-level vs summary; which is now removed per reviewer's request). The categorization of inference procedure generally follows ref. [5]. In the inference procedure, the two-stage estimation procedure comprises two regression stages: the first-stage regression of the exposure on the instrumental variables, and the second-stage regression of the outcome on the fitted values of the exposure from the first stage. Some inference procedures, such as the inverse variance weighted (IVW) procedure (e.g., MR-Egger[8]) or the ratio method (e.g., for SMR[3]) are categorized as two-stage procedure here, as both are asymptotically equivalent to a two-stage estimation procedure in the case of independent instruments. We only list MR methods that directly take input instruments into the model; many MR methods that performs various selection procedures on the instruments (e.g., Guo et al.[21]) are not included. Some recently developed methods that only test for horizontal pleiotropy, such as GLIDE[23] and MR-PRESSO[16] are not included.

as previously observed[29] while LDA MR-Egger produces inflated *p*-values. The poor performance of LDA MR-Egger is presumably due to its fixed effect assumption on **β**, which is not expected to work well in TWAS setting, where the number of SNPs is on the same order of the sample size in gene expression study and where the *cis*-SNPs are all highly correlated with each other due to LD (Supplementary Fig. 2). In the presence of horizontal pleiotropic effects, PMR-Egger becomes the only method that produces calibrated (or slightly conservative) *p*-values (Fig. 1b–d). In contrast, the *p*-values from all other methods become inflated, and more so with increasingly large horizontal pleiotropic effect. For example, when $\gamma$ is $5 \times 10^{-4}$, the genomic control factors from PMR-Egger, SMR, PrediXcan, TWAS, CoMM, and LDA MR-Egger are 0.93, 1.30, 1.33, 1.33, 1.49, and 2.61, respectively. When $\gamma$ is increased to $1 \times 10^{-3}$, the genomic control factors from PMR-Egger, SMR, PrediXcan, TWAS, CoMM, and LDA MR-Egger become 0.93, 2.39, 2.27, 2.46, 4.03, and 2.57 respectively. The null *p*-value distributions from different methods remain largely similar regardless whether the genetic architecture underlying gene expression is sparse or polygenic (Supplementary Fig. 3), regardless of the gene expression heritability (Supplementary Fig. 4), and regardless whether the SNP effects on gene expression are simulated to be correlated with respect to LD or not (Supplementary Fig. 5a, b).

Like MR-Egger, PMR-Egger also makes a relatively strong assumption on pleiotropy that all SNPs have the same horizontal pleiotropic effect. To examine robustness of such assumption, besides the above settings where either 0 or 100% SNPs have horizontal pleiotropic effects, we varied the proportion of horizontal pleiotropic SNPs to be either 10%, 30%, or 50%. We found that PMR-Egger *p*-values remain calibrated regardless of the sparsity of the horizontal pleiotropic SNPs (Supplementary Fig. 6). In addition, besides the above directional pleiotropy settings where the ratio of SNPs with negative vs positive effects is set to be 0:10, we also examined two approximately directional pleiotropy settings (1:9 or 3:7) and one balanced setting (5:5). We found that PMR-Egger *p*-values remain calibrated in either the approximately directional pleiotropy settings or the balanced setting when the horizontal pleiotropic effect is small or moderate ($\gamma = 1 \times 10^{-4}$, $5 \times 10^{-4}$, or $1 \times 10^{-3}$; Supplementary Fig. 7a–c). However, when horizontal pleiotropic effect is large ($\gamma = 2 \times$

$10^{-3}$), as one would expect, PMR-Egger *p*-values become inflated, with genomic control factor being 1.08, 1.31, and 1.37, for settings where the ratio is 1:9, 3:7, and 5:5, respectively (Supplementary Fig. 7d). Cross-gene-based simulations also provide consistent results (Supplementary Figs. 8–12). Although we code genotypes based on allele frequency and kept such coding consistent between simulation and analysis, we found that the results remain consistent when we randomly flip the genotypes of a fraction of SNPs before analysis so that genotype coding does not match between simulation and analysis (Supplementary Fig. 13). Orienting genotypes based on the sign of SNP effects on gene expression[22] in the analysis also yielded largely consistent results with small deflation of *p*-values observable in the presence of large horizontal pleiotropic effects (Supplementary Fig. 14a–c).

Next, we examined the power of different methods to detect non-zero causal effect across various causal effect sizes $\alpha$. Because the same *p*-value from different methods may correspond to different type I errors, we computed power based on false discovery rate (FDR) of 0.1 instead of a nominal *p*-value threshold to allow for fair comparison among methods. When horizontal pleiotropic effects are absent or small, PMR-Egger, TWAS and CoMM have similar power, all outperforming the other three methods, highlighting the importance of making polygenic assumptions on **β** and modeling all *cis*-SNPs together (Fig. 2a, b). The power of PMR-Egger is slightly lower than the other two, presumably because PMR-Egger uses extra parameters to model horizontal pleiotropy, which leads to a loss of degrees of freedom and subsequent loss of power in the absence of horizontal pleiotropy. The power of all methods increases with $\alpha$, though their relative performance rank does not change. In the presence of horizontal pleiotropy, the power of all methods reduces (Fig. 2c, d). However, the power reduction from PMR-Egger is substantially smaller than the other methods. In terms of **β** (Supplementary Fig. 15), we found that the power of different methods in the setting where 10% of SNPs have non-zero effects on gene expression are similar to the baseline setting where all SNPs have non-zero effects, either in the absence (Supplementary Fig. 15e vs Fig. 2a) or presence of horizontal pleiotropic effects (Supplementary Fig. 15f vs Fig. 2d). However, the relative performance of different methods changes when only one SNP or 1% of SNPs have non-zero effect on gene expression. Specifically,

**Table 2 Mean computational time (in s) of various MR methods.**

| Trait | #SNP in the exemplary gene | CoMM | PMR-Egger | TWAS | LDA MR-Egger | SMR | PrediXcan | MR-PRESSO |
|---|---|---|---|---|---|---|---|---|
| T1D from WTCCC (n = 4901) | 300 | 0.51 (0.19) | 0.80 (0.57) | 1.97 (0.86) | 0.08 (0.02) | 0.0003 (0.0005) | 26.74 (2.81) | 408.27 (74.76) |
| | 500 | 1.21 (0.41) | 1.42 (0.77) | 3.48 (1.16) | 0.14 (0.03) | 0.0004 (0.0005) | 11.77 (0.64) | 829.04 (135.79) |
| | 983 | 5.85 (1.50) | 9.79 (1.56) | 4.69 (1.73) | 0.60 (0.09) | 0.0004 (0.0005) | 9.96 (0.78) | 2023.77 (260.43) |
| | 2106 | 111.00 (12.87) | 97.33 (7.63) | 5.87 (2.26) | 4.18 (0.59) | 0.0005 (0.0005) | 22.90 (2.63) | 4913.22 (554.47) |
| Asthma from GERA (n = 61,953) | 300 | 1.47 (0.29) | 2.06 (0.22) | 2.61 (1.48) | 0.05 (0.02) | 0.0002 (0.0004) | 33.39 (3.09) | 464.64 (62.18) |
| | 500 | 1.21 (0.33) | 4.21 (0.81) | 2.54 (0.87) | 0.09 (0.03) | 0.0002 (0.0004) | 11.71 (0.70) | 919.66 (102.83) |
| | 1000 | 24.37 (5.13) | 21.68 (1.66) | 3.07 (2.55) | 0.46 (0.13) | 0.0002 (0.0004) | 14.29 (1.30) | 2275.42 (263.95) |
| | 2008 | 59.01 (4.98) | 52.52 (4.47) | 4.51 (1.48) | 2.33 (0.71) | 0.0004 (0.0005) | 20.18 (3.28) | 5213.73 (601.46) |
| Platelet count from UK Biobank (n = 337,198) | 300 | 2.56 (0.53) | 5.57 (4.54) | 5.04 (4.19) | 0.09 (0.02) | 0.0008 (0.0004) | 10.93 (1.96) | 471.55 (50.44) |
| | 500 | 6.82 (2.75) | 7.61 (2.30) | 5.44 (4.30) | 0.15 (0.02) | 0.0007 (0.0005) | 12.17 (1.04) | 876.06 (92.90) |
| | 1052 | 24.92 (6.28) | 23.59 (3.21) | 5.91 (4.79) | 0.81 (0.09) | 0.0008 (0.0004) | 16.05 (2.38) | 2133.03 (77.56) |
| | 2605 | 186.14 (28.45) | 178.68 (16.75) | 5.37 (0.73) | 8.11 (1.20) | 0.0008 (0.0004) | 9.89 (1.74) | 6949.72 (245.75) |

Computation is carried out on a single thread of a Xeon Gold 6138 CPU. The computational time is averaged across 20 replicates, with values inside parentheses denoting the standard deviation. #SNP denotes the number of cis-SNPs for four exemplary genes in each study. The computational time for MR-PRESSO is based on 10,000 permutations.

in the absence of horizontal pleiotropic effects, the power of both PrediXcan and SMR become slightly higher than PMR-Egger, TWAS and CoMM, all of which have substantially higher power than LDA MR-Egger (Supplementary Fig. 15a, c). The higher power of PrediXcan and SMR in the sparse setting is presumably because the ElasticNet estimation procedure in PrediXcan favors a sparse set of eQTLs while SMR explicitly makes a single eQTL assumption. In the presence of horizontal pleiotropic effects, however, PMR-Egger remains the most powerful, even in the setting where only one SNP has non-zero effect on gene expression (Supplementary Fig. 15b, d). Cross-gene-based simulations also provide consistent results (Supplementary Figs. 16-17). Orienting genotypes based on the sign of SNP effects on gene expression[22] yielded close to zero power, presumably because such approach violates the normality assumption on $\beta$ (Supplementary Fig. 14d).

Finally, PMR-Egger produces accurate estimate of the causal effect $\alpha$, both under the null and under various alternatives, in the presence or absence of horizontal pleiotropic effects (Supplementary Fig. 18), and regardless of the directionality of horizontal pleiotropy (Supplementary Fig. 19a, c, e).

**Simulations: testing and estimating pleiotropic effect.** Our second set of simulations focus on horizontal pleiotropic effect testing. We compared PMR-Egger with LDA MR-Egger and MR-PRESSO. All three methods examine one gene at a time and test whether cis-SNPs within the gene exhibit non-zero horizontal pleiotropic effects.

We first examined type I error control of different methods under the null. We found that PMR-Egger provide calibrated type I error control under a range of causal effect sizes $\alpha$ (Fig. 3). However, p-values from both LDA MR-Egger and MR-PRESSO are inflated, and more so with increasingly large $\alpha$. The overly inflated p-values from LDA MR-Egger is presumably due to its fixed effect modeling assumption on $\beta$ and the subsequent failure to control for realistic LD patterns. The inflation of MR-PRESSO p-values is presumably because MR-PRESSO can only handle independent instruments and thus does not fare well in TWAS settings. Importantly, PMR-Egger p-values remain calibrated regardless of the genetic architecture underlying gene expression (Supplementary Fig. 20) and regardless whether the SNP effects on the gene expression are correlated with respect to LD or not (Supplementary Fig. 5c, d). Cross-gene-based simulations also provide consistent results (Supplementary Figs. 21 and 22).

Next, we examined the power of different methods in detecting non-zero horizontal pleiotropic effect based on an FDR of 0.1. We dropped MR-PRESSO due to its heavy computational burden. We found that the power of PMR-Egger and LDA MR-Egger increases with increasing horizontal pleiotropy, with PMR-Egger outperforming LDA MR-Egger across a range of settings (Fig. 2e, f). The power of both methods is not influenced by the sparsity level of $\beta$ (Supplementary Fig. 23) but depends on the sparsity level of $\gamma$ (Supplementary Fig. 24a). Specifically, power of both methods reduces with increasing sparsity of $\gamma$, though PMR-Egger remains more powerful than LDA MR-Egger across a range of sparsity values. Similarly, the power of both methods to detect pleiotropic effects also suffers in the absence of directional pleiotropic effect (Supplementary Fig. 24b). Cross-gene-based simulations provide consistent results (Supplementary Figs. 25 and 26).

PMR-Egger can estimate the horizontal pleiotropic effect accurately in the presence of directional pleiotropic effect (Supplementary Fig. 27). However, in the absence of directional pleiotropic effect, as expected, the estimates of pleiotropic effects become downward biased, more so in the balanced setting than in

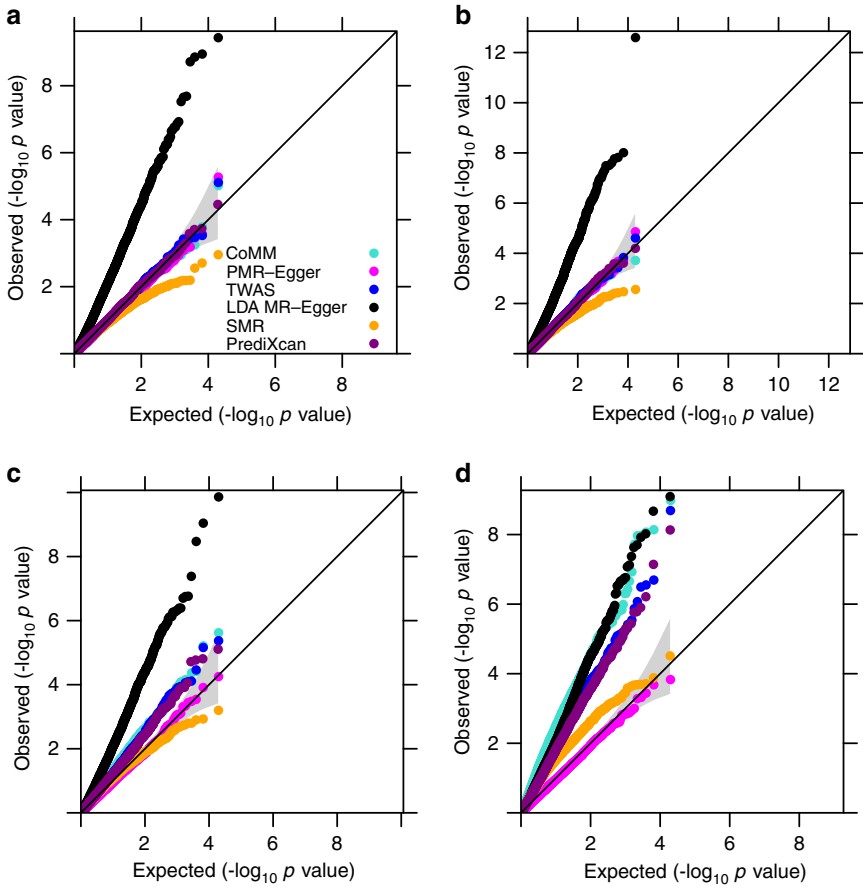

**Fig. 1 Type I error control for different methods on causal effect testing.** Quantile–quantile plot of −log10 p-values from different methods for testing the causal effect either in the absence or in the presence of horizontal pleiotropic effect under null simulations. Compared methods include CoMM (turquoise), PMR-Egger (magenta), TWAS (blue), LDA MR-Egger (black), SMR (orange), and PrediXcan (purple). Null simulations are performed under different horizontal pleiotropic effect sizes: **a** $\gamma = 0$; **b** $\gamma = 0.0001$; **c** $\gamma = 0.0005$; **d** $\gamma = 0.001$. Only p-values from PMR-Egger adhere to the expected diagonal line across a range of horizontal pleiotropic effect sizes.

the approximately directional pleiotropy settings (Supplementary Fig. 19b, d, f).

**Real data applications**. We performed TWAS to detect genes causally associated with each of the 39 phenotypes from three GWASs (details in Methods). The gene expression data are obtained from the GEUVADIS study and contains 15,810 genes. The phenotypes include seven common diseases from Wellcome Trust Case Control study (WTCCC), 22 diseases from Kaiser Permanente/UCSF Genetic Epidemiology Research Study on Adult Health and Aging (GERA), and ten quantitative traits from UK Biobank. The GWAS sample size ranges from 4,686 (for Crohn's disease (CD) in WTCCC) to 337,198 (for UK Biobank). The p-values for testing the causal effect of each gene on the phenotype from different methods are shown for WTCCC traits (Fig. 4a, b; Supplementary Fig. 28), GERA traits (Fig. 5a, b; Supplementary Fig. 29), and UK Biobank traits (Fig. 6a, b; Supplementary Fig. 30); with genomic control factors listed in Supplementary Table 1 and visualized in (Figs. 4c, 5c and 6c). Note that the higher genomic control factor in UK Biobank as compared to WTCCC and GERA is expected under polygenic architecture[30] and reflects at least in part the higher power in the UK Biobank as compared to GERA and WTCCC. While these main analyses use phenotypic residuals after regressing out the effects of top 10 genotype PCs, parallel analysis where the original phenotype was used as the outcome and the top 10 genotype PCs was used as covariates yielded consistent results (Supplementary

Figs. 31–33). For illustration purpose, we display qq-plots for two selected traits in each data, one with a relatively low number of gene associations and the other with a relatively high number of gene associations. Among the selected six traits, the one with zero number of associated genes (bipolar disorder (BD) in WTCCC) and the one with one associated gene (irritable bowel syndrome in GERA), represent approximately null traits with no apparently associated genes. For the six selected traits, consistent with simulations, we found that PMR-Egger p-values are well calibrated, at least more so than the other methods. p-values from CoMM, TWAS, PrediXcan and especially LDA MR-Egger are inflated, while p-values from SMR are overly conservative. The results observed in these exemplary traits generalize to all other examined traits.

We examined the number of associated genes detected by different methods based on a Bonferroni corrected transcriptome-wide threshold (Figs. 4d, 5d and 6d; Supplementary Table 2). The number of detected genes based on this p-value threshold may artificially favor those methods that have inflated type I error control. For this analysis, we excluded LDA MR-Egger for comparison, as its p-values are overly inflated. Consistent with simulations, we found that SMR can barely detect any genes significantly associated with traits across all three data, much less so than that detected by the other four methods. We found that the number of gene-trait pairs detected by CoMM and PMR-Egger is higher than that detected by TWAS and PrediXcan in all three GWASs, again consistent with simulations

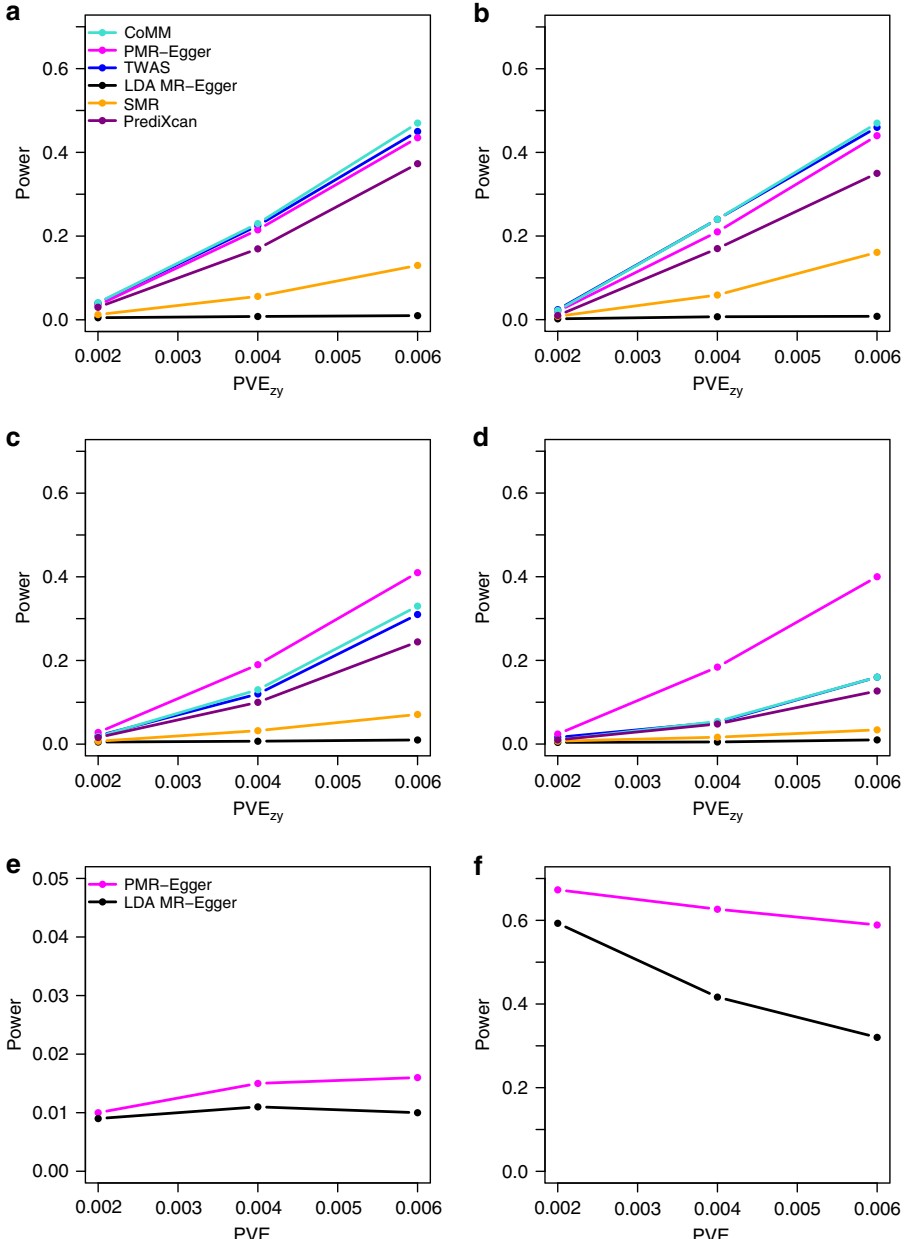

**Fig. 2 Power of different methods under various simulation scenarios.** Power (*y* axis) at a false discovery rate of 0.1 to detect the causal effect **a**–**d** or the horizontal pleiotropic effect **e**, **f** is plotted against different causal effect size characterized by $PVE_{zy}$ (*x* axis). Compared methods include CoMM (turquoise), PMR-Egger (magenta), TWAS (blue), LDA MR-Egger (black), SMR (orange), and PrediXcan (purple). Simulations are performed under different horizontal pleiotropic effect sizes: **a** $\gamma = 0$; **b** $\gamma = 0.0001$; **c**, **e** $\gamma = 0.0005$; **d**, **f** $\gamma = 0.001$.

as well as previous observations that likelihood-based inference often achieves higher power than two-stage inference. However, we do notice that PMR-Egger detects slightly lower number of gene-trait pairs than CoMM based on the same genome-wide *p*-value threshold, consistent with the inflated genomic inflation factors observed for CoMM. Indeed, we found that the estimated $\left|\frac{\alpha}{\gamma}\right|$ for the common set of genes detected by both CoMM and PMR-Egger is higher than the set of genes only detected by CoMM across traits (Supplementary Fig. 34). Therefore, the genes detected by CoMM but not PMR-Egger tend to have large $|\gamma|$ and small $|\alpha|$, likely reflecting false associations due to horizontal pleiotropic confounding.

Overall, by controlling for horizontal pleiotropic effects, PMR-Egger detected many likely causal genes that other methods failed to detect. For example, the *LNK/SH2B3* gene is only identified by

PMR-Egger to be associated with platelet count in the UK Biobank (PMR-Egger $p = 1.17 \times 10^{-221}$; CoMM $p = 0.98$; TWAS $p = 8.6 \times 10^{-5}$; PrediXcan $p = 0.68$; SMR $p = 0.024$). The association between *LNK* and plate count is consistent with results from recent large-scale GWASs[31]. *LNK/SH2B3* encodes the lymphocyte adaptor protein (LNK) that is primarily expressed in hematopoietic and endothelial cells[32]. In hematopoietic cells, LNK functions as a negative regulator of cell proliferation and the thrombopoietin-mediated cytokine signaling pathway, which is a key signaling pathway that promotes megakaryocytes to form platelets[32]. Indeed, platelets are overproduced and accumulated in *Lnk* knockdown cells as well as *Lnk* knockout mouse[33], supporting a causal role of *LNK* in platelets production. As the second example, the *NOD2* gene is identified by PMR-Egger to be associated with Crohn's disease (CD; $p = 6.1 \times 10^{-19}$), and, with a

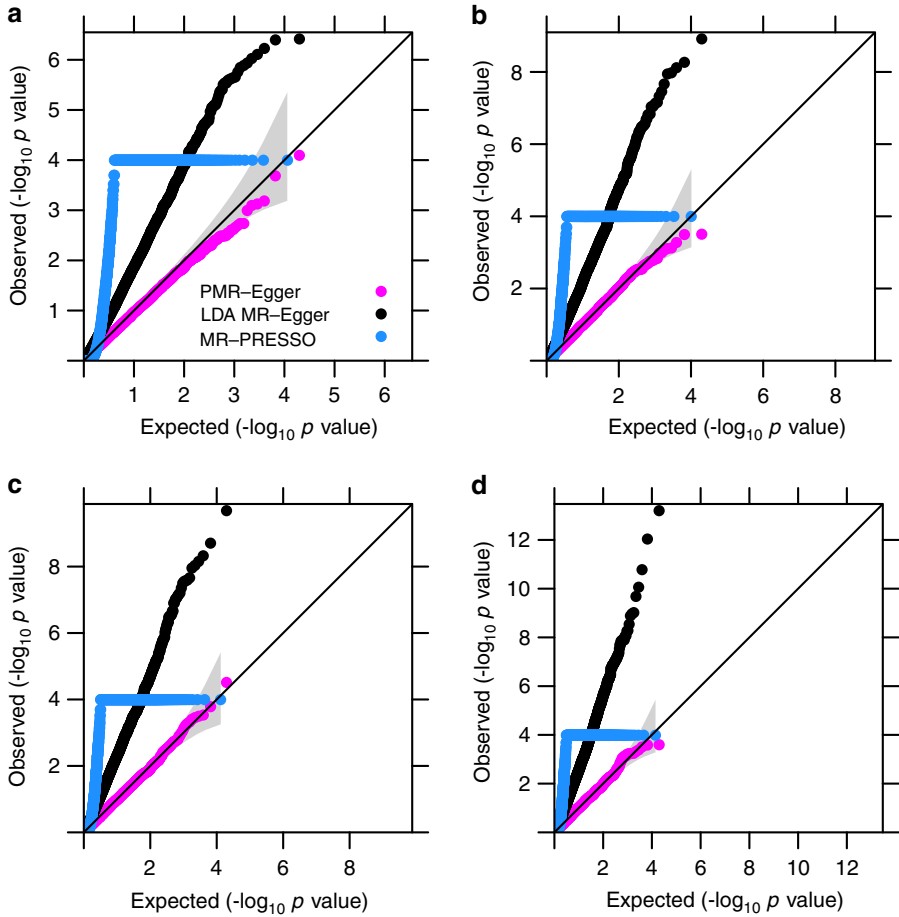

**Fig. 3 Type I error control for different methods on testing pleiotropy.** Quantile–quantile plot of −log10 $p$-values from different methods for testing the horizontal pleiotropic effect either in the absence or in the presence of causal effect under null simulations. Compared methods include PMR-Egger (magenta), LDA MR-Egger (black), and MR-PRESSO (dodger blue). Null simulations are performed under different causal effect sizes characterized by $PVE_{zy}$: **a** $PVE_{zy} = 0$; **b** $PVE_{zy} = 0.2\%$; **c** $PVE_{zy} = 0.4\%$; and **d** $PVE_{zy} = 0.6\%$. Only $p$-values from PMR-Egger adhere to the expected diagonal line across a range of horizontal pleiotropic effect sizes. Due to heavy computational burden, we are only able to run 10,000 permutations for MR-PRESSO. Therefore, the minimal $p$-value from MR-PRESSO is $10^{-4}$.

slightly less significance, also by CoMM ($p = 7.8 \times 10^{-15}$). The association between *NOD2* and CD was not identified by the other methods (TWAS $p = 0.005$; PrediXcan $p = 0.92$; SMR $p = 0.15$). *NOD2* encodes a cytosolic pattern recognition receptor that acts both as a cytoplasmic sensor of microbial products and as an important mediator of innate immunity and inflammatory response[34]. The *NOD2* gene is a well-known susceptible gene for CD and is perhaps one of the first genes ever implied for CD. Multiple SNPs in *NOD2* have been found to be associated with CD in both early linkage studies[35] and many recent GWASs[36]. *NOD2* variants associated with CD often reside in the ligand recognition domain of *NOD2* and can lead to aberrant bacterial handling and antigen presentation[37]. Indeed, *Nod2*-deficient mice displays dysregulated bacterial community in the ileum and *Nod2*-deficient ileal epithelia exhibit impaired ability of inducing immune responses for bacteria elimination[38]. It is thus hypothesized that mis-regulation of *NOD2* can causally lead to altered interactions between ileal microbiota and mucosal immunity, resulting in increased disease susceptibility to CD[38]. As a third example, the *TFRC* gene is identified by PMR-Egger to be associated with red blood cell distribution width (RDW) in the UK Biobank ($p = 3.3 \times 10^{-17}$). Such association is not identified by the other methods (CoMM $p = 0.95$; TWAS $p = 0.76$; PrediXcan $p = 0.97$; SMR $p = 0.38$). *TFRC* encodes the classical transferrin receptor that is involved in cellular iron uptake[39].

Multiple SNPs in *TFRC* have been established to be associated with various erythrocyte phenotypes in GWASs[40]. These associated erythrocyte phenotypes include the mean corpuscular hemoglobin (MCH) and mean corpuscular volume (MCV, the average volume of red blood cells) which is directly related to RDW[39,40]. The variants in *TFRC* likely lead to decreased iron availability for red cell precursors, as has been observed in mice deficient in *Tfrc*, thus resulting in a compensatory increase of red blood cell size as measured by RDW[41]. The regional association plots for these three genes are presented in Supplementary Figs. 35–37.

We also compared the results from different MR methods with a recently published TWAS fine-mapping method, FOCUS[42] (analysis details in "Methods" section). Briefly, we follow[42] and focused on independent and non-overlapping genomic regions that harbor at least one genome-wide significant SNP and at least one significant TWAS gene (Supplementary Table 3). Due to the small number of associated genes detected in WTCCC, we focus mainly in GERA and UK Biobank. There, we found that the results from PMR-Egger is largely consistent with that of FOCUS, more so than the other methods (Supplementary Fig. 38).

Next, we shift our focus to testing horizontal pleiotropic effects. The $p$-values for testing the horizontal pleiotropy effect of each gene on phenotype are shown for WTCCC traits (Fig. 4e, f; Supplementary Fig. 28), GERA traits (Fig. 5e, f; Supplementary

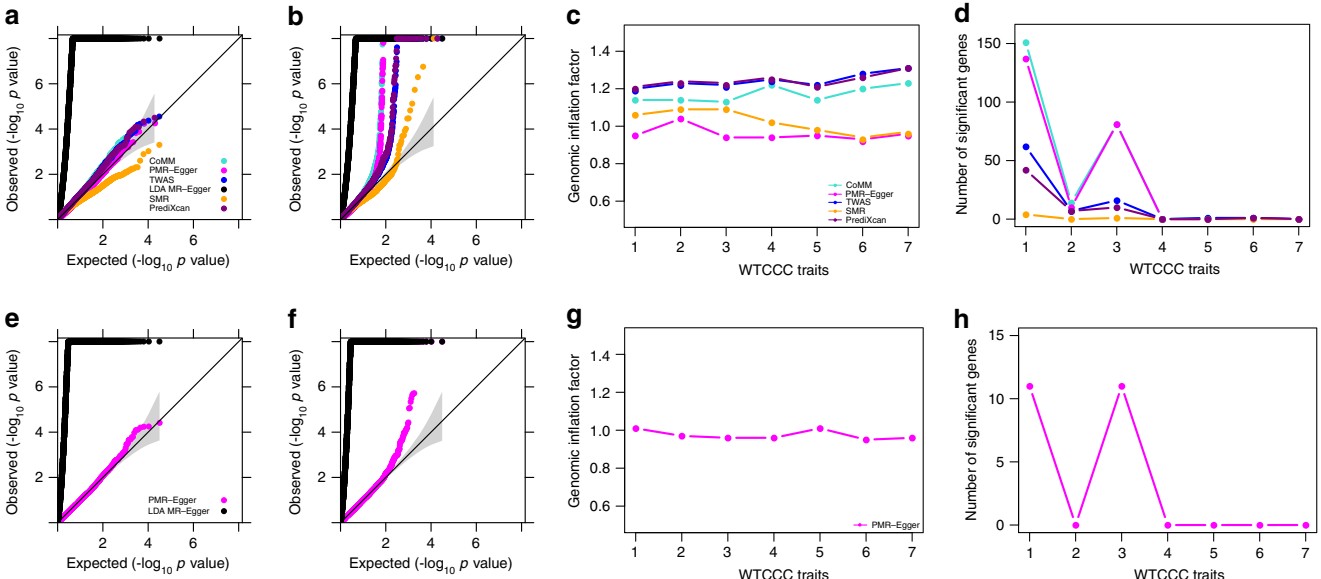

**Fig. 4 TWAS analysis results by different methods for WTCCC traits.** Compared methods include CoMM (turquoise), PMR-Egger (magenta), TWAS (blue), LDA MR-Egger (black), SMR (orange), and PrediXcan (purple). **a** Quantile–quantile plot of −log10 *p*-values from different methods for testing the causal effect for an exemplary trait BD. **b** Quantile–quantile plot of −log10 *p*-values from different methods for testing the causal effect for another exemplary trait T1D. **c** Genomic inflation factor for testing the causal effect for each of the seven traits by different methods. **d** Number of causal genes identified for each of the seven traits by different methods. **e** Quantile–quantile plot of −log10 *p*-values from different methods for testing the horizontal pleiotropic effect for an exemplary trait BD. **f** Quantile–quantile plot of −log10 *p*-values from different methods for testing the horizontal pleiotropic effect for another exemplary trait T1D. **g** Genomic inflation factor for testing the horizontal pleiotropic effect for each of the seven traits by different methods. **h** Number of genes identified to have significant horizontal pleiotropic effect for each of the 7 traits by different methods. For **c**, **d**, **g**, **h**, the number on the *x* axis represents seven traits in order: T1D, CD, RA, BD, T2D, CAD, HT.

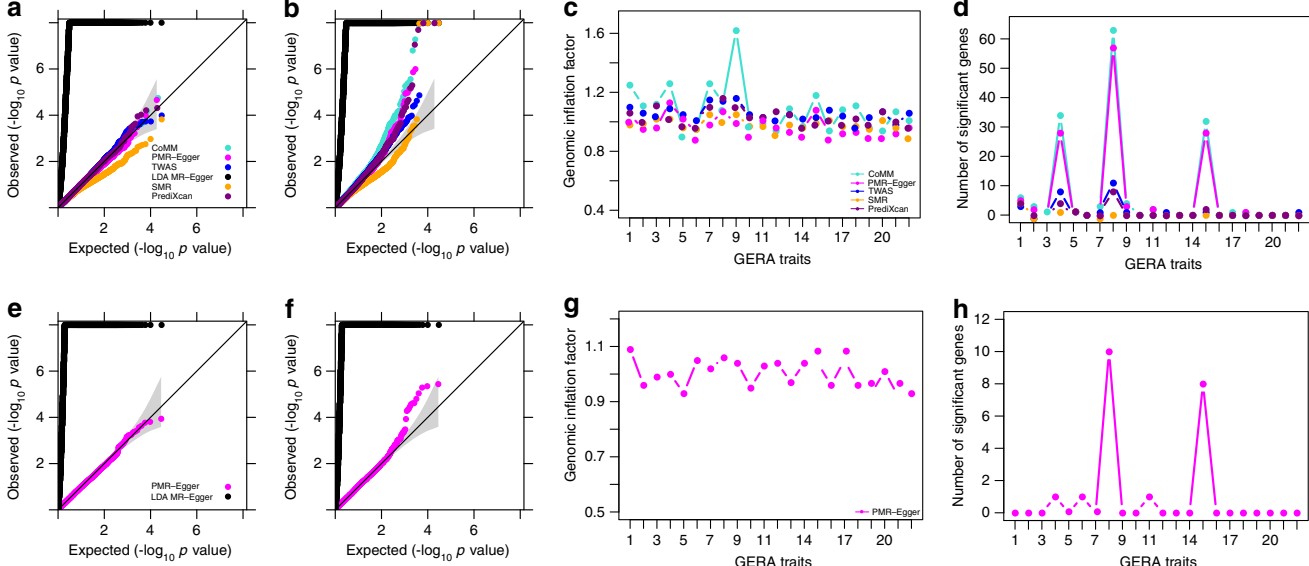

**Fig. 5 TWAS analysis results by different methods for GERA traits.** Compared methods include CoMM (turquoise), PMR-Egger (magenta), TWAS (blue), LDA MR-Egger (black), SMR (orange), and PrediXcan (purple). **a** Quantile–quantile plot of −log10 *p*-values from different methods for testing the causal effect for an exemplary trait irritable bowel syndrome. **b** Quantile–quantile plot of −log10 *p*-values from different methods for testing the causal effect for another exemplary trait asthma. **c** Genomic inflation factor for testing the causal effect for each of the 22 traits by different methods. **d** Number of causal genes identified for each of the 22 traits by different methods. **e** Quantile–quantile plot of −log10 *p*-values from different methods for testing the horizontal pleiotropic effect for an exemplary trait irritable bowel syndrome. **f** Quantile–quantile plot of −log10 *p*-values from different methods for testing the horizontal pleiotropic effect for another exemplary trait Asthma. **g** Genomic inflation factor for testing the horizontal pleiotropic effect for each of the 22 traits by different methods. **h** Number of genes identified to have significant horizontal pleiotropic effect for each of the 22 traits by different methods. For **c**, **d**, **g**, **h**, the number on the *x* axis represents 22 traits in order: asthma, allergic rhinitis, CARD, cancers, depressive disorder, dermatophytosis, T2D, dyslipidemia, HT, hemorrhoids, abdominal hernia, insomnia, iron deficiency, irritable bowel syndrome, macular degeneration, osteoarthritis, osteoporosis, PVD, peptic ulcer, psychiatric disorders, stress disorders, varicose veins.

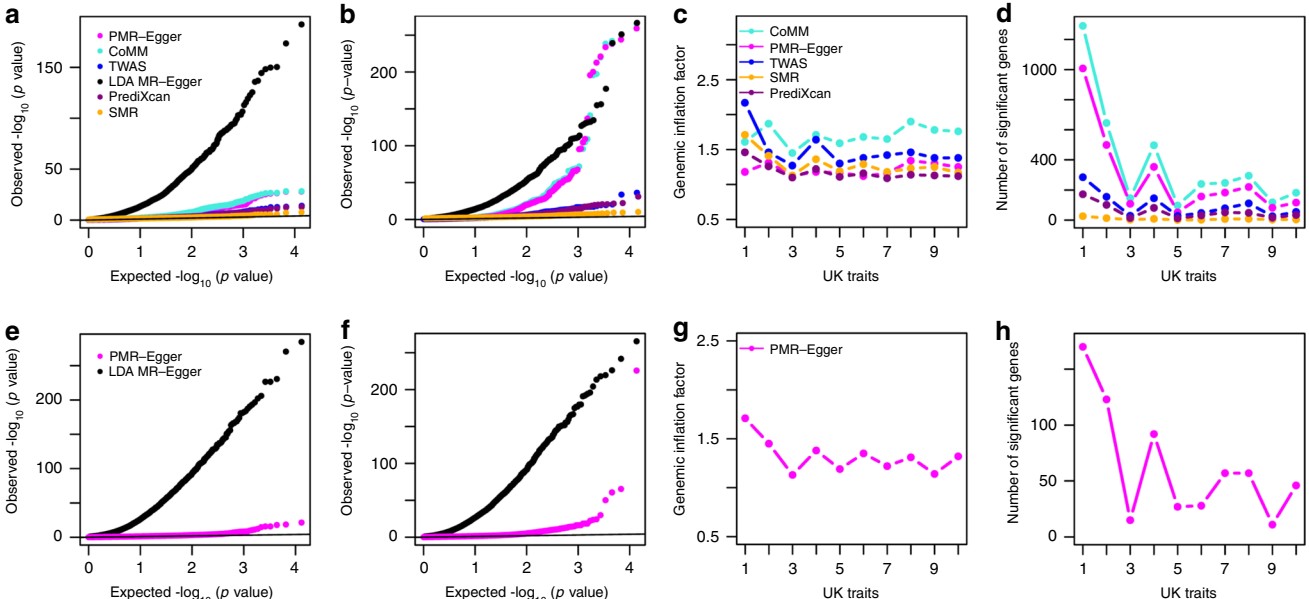

**Fig. 6 TWAS analysis results by different methods for UK Biobank traits.** Compared methods include CoMM (turquoise), PMR-Egger (magenta), TWAS (blue), LDA MR-Egger (black), SMR (orange), and PrediXcan (purple). **a** Quantile–quantile plot of −log10 *p*-values from different methods for testing the causal effect for an exemplary trait BMI. **b** Quantile–quantile plot of −log10 *p*-values from different methods for testing the causal effect for another exemplary trait platelet count. **c** Genomic inflation factor for testing the causal effect for each of the 10 traits by different methods. **d** Number of causal genes identified for each of the 10 traits by different methods. **e** Quantile–quantile plot of −log10 *p*-values from different methods for testing the horizontal pleiotropic effect for an exemplary trait BMI. **f** Quantile–quantile plot of −log10 *p*-values from different methods for testing the horizontal pleiotropic effect for another exemplary trait platelet count. **g** Genomic inflation factor for testing the horizontal pleiotropic effect for each of the 10 traits by different methods. **h** Number of genes identified to have significant horizontal pleiotropic effect for each of the 10 traits by different methods. For **c**, **d**, **g**, **h**, the number on the *x* axis represents 10 traits in order: Height, platelet count, bone mineral density, red blood cell count, FEV1–FVC ratio, BMI, RDW, eosinophils count, forced vital capacity, white blood cell count.

Fig. 39), and UK Biobank traits (Fig. 6e, f; Supplementary Fig. 40); with genomic control factors visualized in (Figs. 4g, 5g, and 6g). We also display qq-plots for the previously selected exemplary traits in (Figs. 4e, f, 5e, f, and 6e, f). Consistent with simulations, *p*-values from PMR-Egger are well behaved while *p*-values from LDA MR-Egger display substantial inflation. For example, the genomic control factor from PMR-Egger ranges from 0.93 to 1.01 in WTCCC, from 0.92 to 1.09 in GERA, and from 1.13 to 1.71 in UK Biobank. In contrast, the genomic control factor from LDA MR-Egger ranges from 34.00 to 36.00 in WTCCC, from 69.82 to 72.19 in GERA and from 17.75 to 29.85 in UK Biobank (Supplementary Table 1). With the same Bonferroni adjusted transcriptome-wide *p*-value threshold, PMR-Egger detected 33 gene-trait pairs in WTCCC in which the *cis*-SNPs exhibit significant horizontal pleiotropy, 37 gene-trait pairs in GERA, and 626 gene-trait pairs in the UK Biobank. The detected horizontal pleiotropic effect tests can help us explain some of the discrepancy in terms of the causal associations detected by PMR-Egger and the other methods (Supplementary Notes).

We note that an important feature of PMR-Egger is its ability to test both causal effect and horizontal pleiotropy effect. We contrast the *p*-values obtained from these two different tests across genes (Supplementary Figs. 41–43). We found that different traits exhibit different gene association patterns. For example, some traits may only contain genes with a significant causal effect but without a significant horizontal pleiotropic effect (e.g., CD and coronary artery disease (CAD) in WTCCC; allergic rhinitis, irritable bowel syndrome and psychiatric disorders in GERA). Some traits may only contain genes with a significant horizontal pleiotropic effect but without a significant causal effect (e.g., dermatophytosis in GERA). Some traits may contain genes with a significant causal effect as well as genes with a significant

horizontal pleiotropic effect, but with the two sets of genes being non-overlapped (e.g., asthma, dyslipidemia, hypertension (HT), abdominal hernia and macular degeneration in GERA; fored vitral capacity in UK Biobank). While the majority of traits contain genes with both a significant causal effect and a significant horizontal pleiotropic effect. Being capable of testing both causal effect and horizontal pleiotropy effect facilitates our understanding of the gene association pattern with various complex traits.

## Discussion

We have presented a data generative model and a likelihood framework for MR analysis that unifies many existing TWAS/MR methods. Under the framework, we have presented PMR-Egger, a new method that conducts MR analysis using multiple correlated instruments while controlling for horizontal pleiotropic effects. By controlling for horizontal pleiotropic effects and making inference under a likelihood framework, PMR-Egger yields calibrated *p*-values across a wide range of scenarios and improves power of MR analysis over existing approaches. Improving the power of TWAS can increase the number of true positives and reduce the number of false positives in the top gene list, potentially leading to better replication and experimental validation on the top identified genes. We have illustrated the benefits of PMR-Egger through extensive simulations and multiple real data applications of TWAS.

One important modeling assumption in PMR-Egger is that the horizontal pleiotropic effects of all SNPs equal. The equal effect assumption directly follows the commonly used Egger assumption for MR analysis and is analogous to the burden assumption commonly used for rare variant tests. Consistent with existing literature on applications of Egger regression and burden test, we

also found that equal effect size assumption employed in PMR-Egger works reasonably robust for causal effect estimation and testing with respect to a range of model mis-specifications and appears to be effective in several real data applications examined here. However, we acknowledge that the equal effect assumption in PMR-Egger can be overly restrictive in many settings. For example, as described in the Results, in the absence of direction pleiotropy, the pleiotropic effect estimate becomes downward biased and the pleiotropic effect test loses power. We have attempted to alleviate this restrictive Egger modeling assumption by imposing an alternative modeling assumption on the horizontal effect sizes based on variance component assumption. Specifically, we have attempted to assume that the horizontal pleiotropic effect of each SNP follows a normal distribution with mean zero and a certain variance component parameter, i.e., analogous to the SKAT assumption[43]. We refer to the resulting model as PMR-VC. Unfortunately, inference for PMR-VC is challenging. Specifically, due to estimation uncertainty in the hyper-parameter estimates, the $p$-values from PMR-VC becomes severely deflated even under simple null simulations (Supplementary Fig. 44). Such deflation of $p$-values has been previously observed in variance component tests for microbiome applications[44]. Only few methods exist to address such $p$-value in-calibration issue resulting from hyper-parameter estimation uncertainty[45], and it is not straightforward to adapt any of these to PMR-VC. In addition, neither PMR-Egger nor PMR-VC can account for correlation between horizontal pleiotropic effects $\gamma$ and SNP effects on gene expression $\beta$. Therefore, while we view PMR-Egger as in important first step towards effective control of horizontal pleiotropic effects in TWAS applications, we emphasize that imposing more realistic modeling assumptions on the horizontal pleiotropic effects in the PMR framework will likely yield more fruitful results in the future.

We have primarily focused on modeling continuous traits with PMR-Egger. For case control studies, we have followed previous approaches and directly treated binary phenotypes as continuous outcomes[14,46–48], which appears to work well in both WTCCC and GERA data applications we examined. Treating binary phenotypes as continuous outcomes can be justified by recognizing the linear model as a first order Taylor approximation to a generalized linear model[46]. However, it would be desirable in the future to extend PMR-Egger to accommodate case control data or other discrete data types in a principled way, by, for example, extending PMR-Egger into the generalized linear model framework.

We have primarily focused on modeling individual-level data with PMR-Egger. However, like many other linear model-based methods in statistical genetics, PMR-Egger can also be easily extended to make use of summary statistics. The summary statistics version of PMR-Egger is described in detail in the Supplementary Notes and implemented in the same software. Briefly, the summary statistics version of PMR-Egger requires marginal SNP effect size estimates and their standard errors, both on the gene expression and on the trait of interest. In addition, it requires a SNP by SNP correlation matrix that can be constructed based on a reference panel. We validated the implementation of the summary statistics-based approach of PMR-Egger through simulations (details in Methods). Specifically, we constructed the SNP by SNP correlation matrix from four different reference panels: all individuals from the GWAS data; 10% randomly selected individuals from GWAS; individuals of European or African ancestry from the 1000 Genomes project. We applied the summary statistics version of PMR-Egger to each reference panel and compared results with the individual-level data-based PMR-Egger that was applied to the complete data. As expected, except in the case when the reference panel is completely unmatched to

the original data (i.e., with the African reference panel), the $p$-values from both approaches for testing causal effects as well as for testing pleiotropy effects are largely consistent with each other, demonstrating the effectiveness of the summary statistics version of PMR-Egger (Supplementary Fig. 45).

Finally, in addition to what we have already mentioned in the Methods, we emphasize again that while we have followed the previous MR literature and use "causal effect" through the text, the effect is causal only when certain MR modeling assumptions hold. These MR assumptions are often not straightforward to validate as it is impossible to measure all confounding factors in the study. Therefore, we caution against the over-interpretation of causal inference in observation studies such as TWAS applications. However, we do believe MR is an important step that allows us to move beyond standard linear regressions and is an important analysis that can provide potentially more trustworthy evidence with regard to causality as compared to simpler regression approaches.

## Methods

**PMR-Egger overview**. We consider a probabilistic Mendelian randomization framework for performing two-sample Mendelian randomization analysis with correlated SNP instruments. Two-sample Mendelian randomization analysis aims to estimate and test for the causal effect of an exposure on an outcome in the setting where the exposure and outcome variables are measured in two separate studies with no sample overlap. In the TWAS applications we consider here, the exposure variable is gene expression level that is measured in a gene expression study, while the outcome variable is a quantitative trait or a dichotomous disease status that is measured in a GWAS. Oftentimes, the gene expression study and GWAS are performed on two separate samples. Although we mostly focus on TWAS applications in the present study, we note that the two-sample Mendelian randomization is also commonly performed in settings where both the exposure and outcome variables are complex traits that are measured in two separate GWASs. An illustrative diagram of MR analysis is displayed in Supplementary Fig. 1.

We denote $\mathbf{x}$ as an $n_1$-vector of exposure variable (i.e., gene expression measurements) that is measured on $n_1$ individuals in the gene expression study and denote $Z_x$ as an $n_1$ by $p$ matrix of genotypes for $p$ instruments (i.e., $cis$-SNPs) in the same study. Note that, unlike standard MR methods that select independent instruments, we follow existing TWAS approaches and use all $cis$-SNPs that are in LD as instruments. We denote $\mathbf{y}$ as an $n_2$-vector of outcome variable (i.e., trait) that is measured on $n_2$ individuals in the GWAS and denote $Z_y$ as an $n_2$ by $p$ matrix of genotypes for the same $p$ instruments there. For both $Z_x$ and $Z_y$, we coded their genotypes based on minor allele count, where the minor allele is defined in the GWAS data. We examined the robustness of such genotype coding in simulations through genotype flipping. We consider three linear regressions to model the two studies separately

$$\mathbf{x} = \boldsymbol{\mu}_x + Z_x\boldsymbol{\beta} + \boldsymbol{\varepsilon}_x, \tag{1}$$

$$\tilde{\mathbf{x}} = \boldsymbol{\mu}_x + Z_y\boldsymbol{\beta} + \boldsymbol{\varepsilon}_{\tilde{x}}, \tag{2}$$

$$\mathbf{y} = \boldsymbol{\mu}_y + \tilde{\mathbf{x}}\alpha + Z_y\boldsymbol{\gamma} + \boldsymbol{\epsilon}, \tag{3}$$

where the Eq. (1) is for the gene expression data and the Eqs. (2) and (3) are for the GWAS data. Here, $\boldsymbol{\mu}_x$ and $\boldsymbol{\mu}_y$ are the intercepts; $\tilde{\mathbf{x}}$ is an unobserved $n_2$-vector of exposure variable on the $n_2$ individuals in the GWAS; $\boldsymbol{\beta}$ is a $p$-vector of instrumental effect sizes on the exposure variable; $\alpha$ is a scalar that represents the causal effect of the exposure variable on the outcome variable; $\boldsymbol{\gamma}$ is a $p$-vector of horizontal pleiotropic effect sizes of $p$ instruments on the outcome variable; $\boldsymbol{\varepsilon}_x$ is an $n_1$-vector of residual error with each element independently and identically distributed from a normal distribution $\mathrm{N}(0, \sigma_x^2)$; $\boldsymbol{\varepsilon}_{\tilde{x}}$ is an $n_2$-vector of residual error with each element independently and identically distributed from the same normal distribution $\mathrm{N}(0, \sigma_x^2)$; and $\epsilon$ is an $n_2$-vector of residual error with each element independently and identically distributed from a normal distribution $\mathrm{N}(0, \sigma_y^2)$. We note that while the above three equations are specified based on two separate studies, they are joined together with the common parameter $\boldsymbol{\beta}$ and the unobserved gene expression measurements $\tilde{\mathbf{x}}$. Equations (2–3) can also be combined into

$$\mathbf{y} = \tilde{\boldsymbol{\mu}}_y + Z_y\boldsymbol{\beta}\alpha + Z_y\boldsymbol{\gamma} + \boldsymbol{\varepsilon}_y, \tag{4}$$

where $\tilde{\boldsymbol{\mu}}_y = \boldsymbol{\mu}_x\alpha + \boldsymbol{\mu}_y, \boldsymbol{\varepsilon}_y = \boldsymbol{\varepsilon}_{\tilde{x}}\alpha + \boldsymbol{\epsilon}$.

Our key parameter of interest in the above joint model is the causal effect $\alpha$. The causal interpretation of $\alpha$ requires two assumptions of MR analysis to hold: (i) instruments are associated with the exposure; (ii) instruments are not associated with any other confounders that may be associated with both exposure and

outcome. Note that our model no longer requires the general exclusion restriction condition of traditional MR (i.e., instruments only influence the outcome through the path of exposure), as we make explicit modeling assumptions on the horizontal pleiotropy effects $\gamma$. Certainly, PMR-Egger still need to satisfy the InSIDE assumption that the instrument-exposure effects and instrument-outcome effects are independent of each other, which is sometimes referred to as the weak exclusion restriction condition[8]. In our model, we derive the causal interpretation and identification of $\alpha$ under the decision-theoretic framework of causal inference[26,49–51] (details in Supplementary Notes). Because the causal effect interpretation of $\alpha$ depends on MR assumptions as well as other explicit modeling assumptions, many of which are not easily testable in practice, MR analysis in observational studies likely provides weaker causality evidence than randomized clinical trials. Therefore, while we follow standard MR analysis and use the term "causal effect" through the text, we only intend to use this term to emphasize the fact that $\alpha$ estimate from an MR analysis is more trustworthy than the effect size estimate in a standard linear regression of $\mathbf{y}$ on $\bar{\mathbf{x}}$.

Because $p$ is often larger than $n_1$, we will need to make additional modeling assumptions on $\beta$ to make the model identifiable. In addition, the two instrumental effect terms defined in Eq. (4), the vertical pleiotropic effect $Z_y \beta \alpha$ and the horizontal pleiotropic effect $Z_y \gamma$, are also not identifiable from each other, unless we make additional modeling assumptions on $\gamma$. Here, we follow standard polygenic model and assume that all elements in $\beta$ are non-zero and that each follows a normal distribution $N\left(0, \sigma_\beta^2\right)$. In addition, we follow the burden test assumption commonly used for rare variant test and assume equal horizontal pleiotropic effects across SNPs $\gamma_j = \gamma$ for $j = 1, \ldots p$. With the burden test assumption on the horizontal pleiotropic effects $\gamma$, our model becomes a generalization of the commonly used MR-Egger regression model. In the special case where instruments are independent and treated as fixed effects and where a two-stage estimation procedure is used for inference, our model reduces to MR-Egger. However, our method can handle general cases where MR-Egger does not apply to. In particular, unlike MR-Egger, our method can handle multiple correlated instruments and perform inference in a likelihood framework.

In the above model, we are interested in estimating the causal effect $\alpha$ and testing the null hypothesis $H_0: \alpha = 0$ in the presence of horizontal pleiotropy effects $\gamma$. In addition, we are interested in estimating the horizontal pleiotropic effect size $\gamma$ and testing the null hypothesis $H_0: \gamma = 0$. We accomplish both tasks through the maximum likelihood inference framework. In particular, we develop an expectation maximization (EM) algorithm for parameter inference by maximizing the joint likelihood defined based on Eqs. (1, 4) (details in Supplementary Notes). The EM algorithm allows us to obtain the maximum likelihood of the joint model, together with maximum likelihood estimates for both $\alpha$ and $\gamma$. In addition, we apply the EM algorithm to two reduced models, one without $\alpha$ and the other without $\gamma$, to obtain the corresponding maximum likelihoods. Afterwards, we perform likelihood ratio tests for either $H_0: \alpha = 0$ or $H_0: \gamma = 0$, by contrasting the maximum likelihood obtained from the joint model to that obtained from each of the two reduced models, respectively. We refer to the above inference procedure as probabilistic, as we place estimation and testing into a maximum likelihood framework. Our inference procedure is in contrast to the commonly used two-stage estimation procedure (as used in, for example, Egger regression[8,22], PrediXcan[1] and TWAS[2]), which estimates $\hat{\beta}$ from Eq. (1) first and then plug in the estimates into Eq. (4) for inference. The previous two-stage estimation procedure fails to properly account for the estimation uncertainty in $\hat{\beta}$ and is known to lose power compared to a formal likelihood inference procedure[5,11,13].

We refer to our model and algorithm together as the two-sample probabilistic Mendelian randomization with Egger regression (PMR-Egger). As explained above, we use "probabilistic" to refer to both the data generative model and the maximum likelihood inference procedure. We use "Egger" to refer to the horizontal pleiotropic assumption on $\gamma$ that effectively generalizes the Egger-regression assumption to correlated instruments. We also note that the joint generative Mendelian randomization model defined in Eqs. (1, 4) is a useful conceptual framework that unifies many existing MR methods. In particular, almost all existing MR methods are built upon the joint model, but with different modeling assumptions on $\beta$ and $\gamma$, and with different inference procedures (Table 1). Compared with these existing MR approaches, PMR-Egger is capable of modeling multiple correlated instruments, effectively controls for horizontal pleiotropy, and places inference into a likelihood framework.

**Simulations.** We performed simulations to assess the performance of PMR-Egger and compare it with existing approaches. To do so, we first obtained 556 cis-SNPs for the gene BACE1 on chromosome 11 from the GEUVADIS data[52] (data processing details in the next section) and simulated gene expression values. We used the gene BACE1 because the number of cis-SNPs in this gene represents the median of all genes. With the scaled genotype data $Z_x$, we simulated SNP effect sizes $\beta$ from a normal distribution $N(0, \text{PVE}_{zx}/556)$, where the scalar $\text{PVE}_{zx}$ represents the proportion of gene expression variance explained by genetic effects. We summed the genetic effects across all cis-SNPs as $Z_x \beta$. In addition, we simulated residual errors $\varepsilon_x$ from a normal distribution $N(0, 1 - \text{PVE}_{zx})$. We then summed the genetic effects and residual errors to yield the simulated gene expression level.

Next, we obtained genotypes for the same 556 SNPs from 2000 randomly selected control individuals in the Kaiser Permanente/UCSF Genetic Epidemiology Research

Study on Adult Health and Aging (GERA)[53,54] and simulated a quantitative trait. Here, we directly used $\beta$ from the gene expression data, which, when paired with the causal effect $\alpha$, yielded the vertical pleiotropic effects $\alpha\beta$. We set $\alpha = \sqrt{\text{PVE}_{zy}/\text{PVE}_{zx}}$, and we simulated residual errors $\varepsilon_y$ from a normal distribution $N(0, 1 - \text{PVE}_{zy})$. Here, the scalar parameter $\text{PVE}_{zy}$ represents the proportion of phenotypic variance explained by vertical pleiotropic effects in the absence of horizontal pleiotropic effects. Afterwards, we simulated horizontal pleiotropic effects $\gamma$ for these SNPs (more details below). We summed the horizontal pleiotropic effects, vertical pleiotropic effects, and residual errors to yield the simulated trait.

In the simulations, we first examined a baseline simulation setting where we set $\text{PVE}_{zx} = 10\%$, $\text{PVE}_{zy} = 0$, with all $\gamma_j = 0$. On top of the baseline setting, we varied one parameter at a time to examine the influence of various parameters. For $\text{PVE}_{zx}$, we set it to be either 1%, 5%, or 10%, close to the median gene expression heritability estimates across genes[55,56]. For $\beta$, we examined alternative SNP effect size distributions that deviate from the polygenic assumption in the baseline setting. Specifically, we randomly selected either 1 SNP, 1%, 10%, or 100% of the SNPs to have non-zero effect, and simulated their effects from a normal distribution to explain a fixed $\text{PVE}_{zx}$ in total. In addition, we examined the case of correlated $\beta$, where the SNP effects on gene expression is generated from a multivariate normal distribution with the covariance matrix $w\Sigma$. Here, $\Sigma$ is the LD matrix among SNPs and $w$ is a scalar that is chosen to ensure that $\text{PVE}_{zx}$ equal to 10%. For $\text{PVE}_{zy}$, we varied its value to be either 0% (for null simulations), 0.2%, 0.4%, or 0.6% (for power simulations). For the horizontal pleiotropy effects $\gamma$, we randomly assigned a fixed proportion of $\gamma_j$ to be non-zero (proportion equals 10%, 30%, 50%, or 100%). Afterwards, we set the absolute value of non-zero $\gamma_j$ to be the same value of $\gamma$. As a sensitivity analysis, we also randomly assigned some of their signs to be positive and some of their signs to be negative, with the ratio of positive effects to negative effects being either 1:9, 3:7, or 5:5. Here, we set $\gamma$ to be $1 \times 10^{-4}$, $5 \times 10^{-4}$, $1 \times 10^{-3}$, or $2 \times 10^{-3}$, which corresponds to the 50%, 70%, 90%, 95% quantiles of horizontal pleiotropic effect estimates across all genes and all traits in the WTCCC data (more details below). While genotype coding is based on allele frequency in both simulations and analysis, we also examined cases where we randomly flipped the genotypes of a fraction of SNPs before analysis so that genotype coding does not match between simulation and analysis. The faction of flipped genotype SNPs is set to be either 10%, 30%, or 50%. In addition, we conducted analysis by orienting SNP genotypes based on its estimated effect sign on the gene expression, to examine whether such "positive orientation" strategy can improve the performance of PMR-Egger. For null simulations and type I error control examination, we performed 10,000 simulation replicates for each simulation scenario described above. For power calculation, for each scenario, we performed 1000 alternative simulations together with 9000 null simulations and calculated power based on false discovery rate (FDR).

While we applied PMR-Egger to analyze individual-level data from all simulations, we also applied PMR-Egger to analyze summary statistics in a subset of simulations to validate the implementation of the summary statistics-based PMR-Egger algorithm. These results are presented in the Discussion section. Here, we considered the simulation settings with a fixed sample size ($n_1 = 465$, $n_2 = 2000$), different causal effect sizes ($\text{PVE}_{zy} = 0$ or 0.6%) and different pleiotropy effect sizes ($\gamma = 0$ or 0.0005). In the analysis, we calculated the LD matrix in the eQTL data using the observed individual-level genotypes in the eQTL study. We calculated the LD matrix in the GWAS data from a reference panel. The reference panel is constructed in four different ways, by using individual-level genotypes from either all individuals in the GWAS ($n = 2000$), 10% of randomly selected individuals from the GWAS ($n = 200$), individuals with European ancestry ($n = 503$) or individuals with African ancestry ($n = 611$) from the 1000 Genomes project phase 3. Note that the African ancestry panel includes 99 Esan in Nigeria (ESN), 113 Gambian in Western Division, Mandinka (GWD), 99 Luhya in Webuye, Kenya (LWK), 85 Mende in Sierra Leone (MSL), 108 Yoruba in Ibadan, Nigeria (YRI), 96 African Caribbean in Barbados (ACB), and 61 people with African Ancestry in Southwest USA (ASW).

Besides the single gene-based simulations, we also conducted cross-gene simulations. Specifically, we randomly selected 10,000 genes from GEUVADIS. We extracted cis-SNPs for these 10,000 genes, obtaining a median of 576 cis-SNPs per gene (min = 11; max = 7409). For each gene in turn, we used its cis-SNPs to simulate its gene expression level as described above. Afterwards, we applied different methods to analyze simulated data. The cross-gene-based simulations reflect the varying LD pattern and the varying number of cis-SNPs across genes that we observe in real data, and thus are likely to be more realistic than the single gene-based simulations. We performed cross-gene simulations under all simulation settings described above, including settings with varying gene expression heritability, varying genetic architectures underlying gene expression, as well as varying causal and horizontal pleiotropy effects.

**Real data applications.** We applied our method to perform TWAS by integrating gene expression data with several GWASs. Specifically, we obtained GEUVADIS data[52] as the gene expression data and examined 39 phenotypes from three GWASs. The three GWASs include the Wellcome Trust Case Control study (WTCCC)[57], the Kaiser Permanente/UCSF Genetic Epidemiology Research Study on Adult Health and Aging (GERA)[53,54], and the UK Biobank[58].

The GEUVADIS data[52] contains gene expression measurements for 465 individuals collected from five different populations that include CEPH (CEU), Finns (FIN), British (GBR), Toscani (TSI) and Yoruba (YRI). In the expression data, we only focused on protein coding genes and lincRNAs that are annotated in GENCODE (release 12)[59,60]. Among these genes, we removed lowly expressed genes that have zero counts in at least half of the individuals to obtain a final set of 15,810 genes. We performed PEER normalization to remove confounding effects and unwanted variations following previous studies[14,61]. Afterwards, following[14], to remove remaining population stratification, we quantile normalized the gene expression measurements across individuals in each population to a standard normal distribution, and then further quantile normalized the gene expression measurements to a standard normal distribution across individuals from all five populations. Besides expression data, all individuals in GEUVADIS also have their genotypes sequenced in the 1000 Genomes Project. We obtained genotype data from the 1000 Genomes Project phase 3. We filtered out SNPs that have a Hardy-Weinberg equilibrium (HWE) $p$-value $< 10^{-4}$, a genotype call rate $<95\%$, or a minor allele frequency (MAF) $<0.01$. We retained a total of 7,072,917 SNPs for analysis.

The WTCCC data consists of about 14,000 cases from seven common diseases and 2938 shared controls[57]. The diseases include type 1 diabetes (T1D; $n = 1963$), Crohn's disease (CD; $n = 1748$), rheumatoid arthritis (RA; $n = 1861$), bipolar disorder (BD; $n = 1868$), type 2 diabetes (T2D; $n = 1924$), coronary artery disease (CAD; $n = 1926$), and hypertension (HT; $n = 1952$). We obtained quality controlled genotypes from WTCCC and initially imputed missing genotypes using BIMBAM[62] to arrive at a total of 458,868 SNPs shared across all individuals. Afterwards, we further imputed SNPs using the 1000 Genomes as the reference panel using SHAPEIT and IMPUTE2[63]. We filtered out SNPs that have an HWE $p$-value $< 10^{-4}$, a genotype call rate $<95\%$, or an MAF $< 0.01$ to obtain a total of 2,793,818 imputed SNPs. For each trait in turn, we first regressed the phenotype on the top 10 genotype principal components (PCs) and obtained phenotype residuals. We then scaled the phenotype residuals to have a mean of zero and standard deviation of one and used these phenotype residuals for TWAS analysis. In addition to the main analysis that uses phenotype residuals, we also performed parallel analysis with PMR-Egger where we used the original phenotype as the outcome variable and the top 10 genotype PCs as covariates.

The GERA study consists of 61,953 individuals and 675,367 genotyped SNPs. We filtered out SNPs that had a genotype calling rate below 0.95, MAF < 0.01, or HWE $p$-value $< 10^{-4}$ to yield a total of 487,609 SNPs. We phased genotypes using SHAPEIT[64] and imputed SNPs based on the Haplotype Reference Consortium (HRC version r1.1) reference panel[65] on the Michigan Imputation Server using Minimac3[66]. Afterwards, we further filtered out SNPs that have a HWE $p$-value $< 10^{-4}$, a genotype call rate $<95\%$, an MAF $< 0.01$, or an imputation score $< 0.30$ to arrive at a total of 8,385,867 SNPs that are shared across 61,953 individuals. We examined 22 diseases in GERA that include asthma (number of cases $n = 10,101$), allergic rhinitis ($n = 15,193$), cardiovascular disease (CARD, $n = 16,431$), cancers ($n = 18,714$), depressive disorder ($n = 7900$), dermatophytosis ($n = 8443$), type 2 diabetes (T2D, $n = 7638$), dyslipidemia ($n = 33,071$), hypertension (HT, $n = 31,044$), hemorrhoids ($n = 9922$), abdominal hernia ($n = 6876$), insomnia ($n = 4357$), iron deficiency ($n = 2706$), irritable bowel syndrome ($n = 3367$), macular degeneration ($n = 4031$), osteoarthritis ($n = 22,062$), osteoporosis ($n = 5909$), peripheral vascular disease (PVD, $n = 4718$), peptic ulcer ($n = 1007$), psychiatric disorders ($n = 9408$), stress disorders ($n = 4706$), and varicose veins ($n = 2714$). For each trait in turn, we first regressed the phenotype on the top 10 genotype principal components (PCs) and obtained phenotype residuals. We then scaled the phenotype residuals to have a mean of zero and standard deviation of one and used these phenotype residuals for TWAS analysis. In addition to the main analysis that uses phenotype residuals, we also performed parallel analysis with PMR-Egger where we used the original phenotype as the outcome and the top 10 genotype PCs as covariates.

The UK Biobank data consists of 487,409 individuals and 92,693,895 imputed SNPs[58]. We followed the same sample QC procedure in Neale lab (https://github.com/Nealelab/UK_Biobank_GWAS/tree/master/imputed-v2-gwas) to retain a total of 337,198 individuals of European ancestry. We filtered out SNPs with an HWE $p$-value $< 10^{-7}$, a genotype call rate $<95\%$, or an MAF $< 0.001$ to obtain a total of 13,876,958 SNPs. We selected 10 UK Biobank quantitative traits that have a phenotyping rate $>80\%$, a SNP heritability $> 0.2$ and a low correlation among them following a previous study[67]. The 10 traits include height ($h^2 = 0.579$;), Platelet count ($h^2 = 0.404$), bone mineral density ($h^2 = 0.401$), red blood cell count ($h^2 = 0.324$), FEV1–FVC ratio ($h^2 = 0.313$), body mass index (BMI, $h^2 = 0.308$), RBC distribution width ($h^2 = 0.288$), Eosinophils count ($h^2 = 0.277$), forced vital capacity ($h^2 = 0.277$), white blood cell count ($h^2 = 0.272$). For each trait in turn, we regressed the resulting standardized phenotypes on sex and top 10 genotype principal components (PCs) to obtain the residuals, standardized the residuals to have a mean of zero and a standard deviation of one, and finally used these scaled residuals to conduct TWAS analysis. We also performed parallel analysis with PMR-Egger by using the original phenotype and including the top 10 genotype PCs as covariates.

We combined the GEUVADIS data with each of the three GWASs for TWAS analysis. To do so, in the GEUVADIS data, for each gene in turn, we extracted *cis*-SNPs that are within either 100 kb upstream of the transcription start site (TSS) or 100 kb downstream of the transcription end site (TES). We overlapped these SNPs

in GEUVADIS with the SNPs obtained from each of the three GWASs to obtain common sets of SNPs. The median number of the overlapped *cis*-SNPs between GEUVADIS and WTCCC, GERA or UK Biobank are 200, 556, or 500, respectively. Afterwards, for each pair of gene (from GEUVADIS) and trait (from GWAS) in turn, we examined the causal relationship between gene expression and trait of interest while testing and controlling for potential horizontal pleiotropic effects.

**Compared methods.** For testing the causal effect, we compared the performance of PMR-Egger with five existing methods that include: (1) SMR, which uses a single instrument and does not control for horizontal pleiotropy. For SMR, we first performed a linear regression to choose the top associated *cis*-SNP to be the instrumental variable. (2) PrediXcan, which uses multiple correlated instruments but does not control for horizontal pleiotropy. For PrediXcan, we used all *cis*-SNPs for the model and used ElasticNet implemented in the R package glmnet to obtain the coefficient estimates for the *cis*-SNPs. (3) TWAS, which uses multiple correlated instruments but does not control for horizontal pleiotropy. For TWAS, we used all *cis*-SNPs for the model and used BSLMM[46] implemented in the GEMMA software[68] to obtain coefficient estimates for the *cis*-SNPs. (4) CoMM, which uses multiple correlated instruments but does not control for horizontal pleiotropy. We used all *cis*-SNPs for the model and used the R package CoMM for model fitting. (5) LDA MR-Egger, which uses multiple correlated instruments and controls for horizontal pleiotropy. We used all *cis*-SNPs for the model and contacted the authors of LDA MR-Egger to obtain the method source code. All these methods are suitable for two-sample design and yield $p$-values for testing the causal effect $\alpha$. Note that PrediXcan, TWAS and CoMM are not originally described as an MR method but conceptually rely on the same joint MR model based on Eqs. (1) and (4). These three methods differ in their prior assumptions on $\boldsymbol{\beta}$: PrediXcan relies on ElasticNet assumption; TWAS relies on BSLMM[46] assumption; whereas CoMM relies on the normal prior assumption. In addition, PrediXcan and TWAS rely on a two-stage regression procedure while CoMM is based on maximum likelihood. Also, while the prior used in PrediXcan is polygenic, the parameter estimates obtained from PrediXcan is sparse as it uses posterior mode instead of posterior mean. We were unable to compare our method with either GSMR or the standard Egger regression, as both require multiple independent SNP instruments that are generally not feasible to obtain in TWAS applications.

Again, we used all *cis*-SNPs for methods that can make use of multiple correlated instruments (i.e., PMR-Egger, TWAS, PrediXcan, CoMM, and LDA MR-Egger). We performed a linear regression to select the top associated *cis*-SNP as the instrumental variable for SMR, as it can only use a single instrument. In all simulations and real data applications, methods that can use either individual-level data or summary statistics (PMR-Egger, PrediXcan, and TWAS) are applied using individual-level data as input to ensure their optimal performance. Methods that can only use individual-level data (CoMM) are applied using individual-level data as input. Methods that can only use summary statistics (SMR and LDA MR-Egger) are applied using summary data as input. For PMR-Egger, we used individual-level data for all main analyses and used summary data for a subset of analyses that are described in the Discussion section.

Besides the above methods, we also compared different methods to a recently published fine-mapping TWAS method, FOCUS[42]. In the FOCUS analysis, we followed[42] and obtained a set of independent non-overlapping genomic regions termed as LD blocks from LDetect[69]. We removed genomic regions that overlap with the MHC region due to the extensive LD structure there. Following[42], we also focus our analysis on a subset of regions that harbor at least one genome-wide-significant SNP ($p < 5 \times 10^{-8}$; the default threshold used in FOCUS), and for each TWAS/MR method (i.e., PMR-Egger, TWAS, PrediXcan, CoMM, or SMR), also harbor at least one TWAS gene that is declared significant by the given method. We then applied FOCUS to analyze these remaining regions and identify genes that are in the 90% credible set.

For testing horizontal pleiotropic effect, we compared the performance of PMR-Egger with two existing methods that include (1) LDA MR-Egger; and (2) the global test in MR-PRESSO, which is implemented as an R package. Both these methods examine one gene at a time and output a $p$-value for testing horizontal pleiotropic effects. Note that, unlike PMR-Egger and LDA MR-Egger, MR-PRESSO requires independent instruments and uses permutation to obtain the empirical $p$-values. Due to the heavy computational burden resulting from permutations, we restricted the number of permutations in MR-PRESSO to 10,000 (the lowest possible $p$-value from MR-PRESSO is thus $10^{-4}$) and were only able to apply MR-PRESSO to a subset of simulation scenarios.

**Reporting summary.** Further information on research design is available in the Nature Research Reporting Summary linked to this article.

## Data availability

No data were generated in the present study. The GEUVADIS gene expression data are publicly available at http://www.geuvadis.org. The WTCCC genotype and phenotype data are publicly available at https://www.wtccc.org.uk. The GERA genotype and phenotype data are available at https://www.ncbi.nlm.nih.gov/gap with dbGaP accession number phs000788. The UK Biobank data are from UK Biobank resource at https://mrc.ukri.org/research/facilities-and-resources-for-researchers/biobank/.

## Code availability

Our method is implemented in the R package PMR, freely available at http://www.xzlab.org/software.html and https://github.com/yuanzhongshang/PMR. The code to reproduce all the analyses are available on GitHub https://github.com/yuanzhongshang/PMRreproduce.

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

## Acknowledgements

This study was supported by the National Institutes of Health (NIH) Grants R01HG009124 and R01HL142023, and National Science Foundation (NSF) Grant DMS1712933. Z.Y. is currently supported by the National Natural Science Foundation of China (81872712 and 81673272), the Natural Science Foundation of Shandong Province (ZR2019ZD02) and the Young Scholars Program of Shandong University (2016WLJH23) after moving back to China. We thank the Wellcome Trust Centre for Human Genetics for making the heterogenous stock mouse data available online. This study makes use of data generated by the Wellcome Trust Case Control Consortium (WTCCC). A full list of the investigators who contributed to the generation of the data are available from https://www.wtccc.org.uk/. Funding for the WTCCC project was provided by the Wellcome Trust under award 076113 and 085475. The GERA Data (dbGaP accession number phs000788) came from a grant, the Resource for Genetic Epidemiology Research in Adult Health and Aging (RC2 AG033067; Schaefer and Risch, PIs) awarded to the Kaiser Permanente Research Program on Genes, Environment, and Health (RPGEH) and the UCSF Institute for Human Genetics. The RPGEH was supported by grants from the Robert Wood Johnson Foundation, the Wayne and Gladys Valley Foundation, the Ellison Medical Foundation, Kaiser Permanente Northern California, and the Kaiser Permanente National and Northern California Community Benefit Programs. The RPGEH and the Resource for Genetic Epidemiology Research in Adult Health and Aging are described in ref. [70]. This study has been conducted using UK Biobank resource under Application Number 30686. UK Biobank was established by the Wellcome Trust medical charity, Medical Research Council, Department of Health, Scottish Government and the Northwest Regional Development Agency. It has also had funding from the Welsh Assembly Government, British Heart Foundation and Diabetes UK.

## Author contributions

X.Z. conceived the idea and provided funding support. X.Z. and Z.Y. developed the methods. Z.Y. developed the software tool with assistance from J.L. and C.Y. Z.Y. performed simulations and real data analysis with assistance from H.Z., P.Z., S.Y., and S.S. X.Z. and Z.Y. wrote the manuscript with input from all other authors. All authors reviewed and approved the final manuscript.

## Competing interests

The authors declare no competing interests.
