## [Peer Review File · Nature Communications]

Reviewers' comments:

Reviewer #1 (Remarks to the Author):

Overview

=====

The manuscript describes an approach for identifying a causal effect of gene expression on a complex trait using a likelihood framework called PMR-Egger.

The assumptions underlying the likelihood framework is relaxed compared with previous (strict) MR approaches in that it allows for a particular direct effect of SNPs on trait (ie single shared horizontal pleiotropic effect). The approach is sound and unifies various approaches in a single statistical model.

The authors demonstrate that PRM-Egger is unbiased under various simulations and apply it to GWAS data from WTCCC, GERA, and UKBB studies. I think the approach is quite nice and the simulations are thorough; however, I do have some comments that should be addressed.

Main comments

=====

1. Across all simulations the authors used a single gene, BACE1, as a basis for comparison. There is some justification for why this gene was selected, but I think it would be much more informative to see the performance of PMR-Egger (and its comparison) across genes selected at random. The reason being that LD patterns vary dramatically and I worry that results are not reflective of how it will be used in real data.
2. It isn't clear to me if other methods are compared using respective individual-level approach or summary-based approaches. This should be much more explicit (both for simulations and real data).
3. Why are results for LDA MR-Egger so extreme? LDA MR Egger is essentially TWAS but with an added intercept, which should reduce overall power. Based on my reading it also estimates an additional variance term compared with standard TWAS, which should further reduce its power in comparison. I understand that the authors did not design MR Egger, but the results seem more like a bug.
4. The authors perform nice simulations demonstrating unbiased estimates of the pleiotropic effect under various perturbations to model assumptions. Can the authors perform a set of simulations that randomly flips the encoding at a subset of genotypes prior to the estimation procedure. It is not clear how the approach can know a-priori what the correct encoding is to estimate the directional effect over all data.
5. In real data, authors should include genotyping PCs to minimize the impact of ancestry/stratification. While this doesn't fit the model as explicitly defined (single μ_x) it should be estimable and adjusted for under a fixed-effect model ($\mu_x = X\beta$) as is standard in other approaches.
6. Why does TWAS have such a large average (and std) time for its first entry in Table 1? Is this the result of a single outlier?
7. Genomic control estimates reflect differential power in WTCCC/GERA vs UKBB studies

8. The manuscript describes several examples where associations go away after controlling for nearby genes as evidence in support of pleiotropic effects. An alternative and simpler explanation for these examples is that SNPs used in the model are simply tagging nearby eQTLs (or some molecular feature) rather than directly impacting downstream trait through other pathways. This approach is modeled in a recent method FOCUS.

9. FOCUS is a recently published tool that attempts to control for pleiotropic effects in TWAS by jointly modelling multiple genes at a GWAS region. Can the authors compare, at a minimum, real-data results of PMR-Egger with FOCUS?

10. The authors describe a very nice extension of their approach to estimate the causal (and pleiotropic) effect from summary data. Considering the summary based approach will likely be the most common use-case scenario it is important that the authors demonstrate its performance compared with the individual-level approach.

Minor comments

=====

1. I would like to commend the authors for including their source code along with the manuscript.

Reviewer #2 (Remarks to the Author):

Thank you for the opportunity to review this manuscript. The authors have put together a method with the aim of distinguishing causality from horizontal pleiotropy in transcriptome-wide association studies. Whilst this is an area in the field where methods development is needed, I'm afraid that I have several concerns regarding this approach. This in particular is highlighted by the real data examples chosen in the study, amongst which the authors suggest their method implicates HERPUD1 as the causal gene responsible for an association signal with dyslipidemia on chromosome 16. This gene neighbors CETP, perhaps the most well-known of lipid gene to date. I have attached my more detailed comments below in case the authors find them useful in refining their manuscript.

Major comments:

Page 6 – The authors suggest that Cochran's Q can test for horizontal pleiotropy, although this is misleading. This test is commonly used to assess heterogeneity in 2-sample MR analyses, however the presence of heterogeneity does not mean that a causal relationship does not underly an association. I recommend that this should be removed.

Page 8 – At the start of the methods section the authors refer to gene expression as an 'explanatory variable' in their analysis. I assume they mean it's being treated as their 'exposure' in an MR setting?

Page 10 – My main concern with this method is that it relies on an assumption used in rare variant analyses (i.e. the burden test method) that variants are independent. The authors state that under this assumption their model 'reduces to the commonly used MR-Egger regression model', which of course relies of the assumption of independent instruments. Moreover, in a conventional MR setting the MR-Egger approach only becomes powerful in terms of distinguishing horizontal pleiotropy when many (i.e. dozens) of independent genome-wide (strong) instruments are used. This is of course incredibly unrealistic for cis-eQTL, particularly using a sample size of n=465.

I recommend that the authors test this implicitly in their paper i.e. simulating a correlation structure amongst eQTL and assessing whether false positive rates increase in the presence of

higher correlation. If it does then perhaps incorporating a reference panel, similar to conventional 2-sample MR analyses, may assist in removing tightly correlated instruments that may bias findings.

Page 26 – In the real data application of their method, the authors state that ‘Overall, by controlling for horizontal pleiotropic effects, PMR-Egger detected many likely causal genes that the other methods failed to detect.’ Conventionally these sections of papers concern cherry-picked examples where their method outperforms competitors. However, I find the examples quite concerning.

The first 2 examples are both in the HLA region of the genome (C2 & HLA-F), which is a region of the genome many previous approaches do not attempt to investigate due to the extensive LD structure here making disentangling causal genes extremely problematic. I am therefore very skeptical that these genes are likely the causal genes for association signals in this region, particularly giving concerns regarding independent eQTL not being accounted for. For example, did the authors assess whether eQTL for these genes in their dataset are also eQTL for various neighboring genes? My guess is that there would be a lot of genes where this is the case. I would strongly recommend choosing genes outside of HLA to showcase your method.

Even more alarming is the third example – which is HERPUD1 and its association with dyslipidemia. The authors state this is a ‘well-known lipid-related gene’ and cite 2 genetic association studies. I was unable however to find this gene symbol in either study. It may be tucked away in the supplementary material somewhere. However, I stopped looking when discovering that this gene neighbors CETP, which is actually a well-known lipids gene. The association signal at the locus is therefore most likely due to CETP, and HERPUD1 I imagine is simply co-expressed with it. I would again strongly encourage the authors to take this example out of their manuscript and consider alternatives. Again, across ~20,000 coding genes some cherry-picking to best showcase a method is not uncommon in these types of studies. However, this makes the choice of these 3 examples slightly worrying.

Minor comments:

Page 5 – ‘multiple independent/near-independent SNP instruments’ – strictly instrumental variables for established 2-sample MR approaches should be independent – it’s only how one defines ‘independent’ that allows ‘near-independent’ SNPs to creep into analyses

Page 5 – minor typo ‘suggest’ should be ‘suggests’ on last line

Page 6 – ‘Cohran’s’ should be ‘Cochran’s’

Some of the grammar and language is slightly hard to follow throughout.

Reviewer #3 (Remarks to the Author):

The authors presented the developed method PMR-Egger, estimating both causal and pleiotropic effects simultaneously. PMR-Egger follows the framework and the assumptions of MR-Egger, but using likelihood test to do the tests and estimate the effects. The simulations showed PMR-Egger was well calibrated and had larger power than existing methods. The real data analysis showed more genes were detected by PMR-Egger. However, the underlying biological model is not clear to me. Therefore, I am not convinced about the results.

Major comments

1. The underlying biological model is unclear. From my understanding, there are not many causal

variants that have effects on gene expressions. The PMR-Egger analyses were done under the assumption of polygenic model, which is opposite to what I thought. Moreover, they assumed many cis-SNPs may have pleiotropic effects, which I do not understand.

1.1 We assume there are few causal variants, i.e. only a few top SNPs can capture most of the genetic variation. SMR assumes there is a single causal variant, selecting the top cis-SNP. In practice, the top cis-SNP may not capture all the genetic variation. Therefore, other methods, eg., "TWAS" or "PrediXcan" has larger power. I don't think using all the cis-SNPs may gain much extra power. Even in the simulation where many causal variants were simulated, PMR-Egger does not show larger power under the causality model (Fig 2a).

1.2 From my understanding, when there is a single causal variant, the causal variant has either causal ($z \rightarrow x \rightarrow y$) or pleiotropic effect ($z \rightarrow x, z \rightarrow y$). z represents SNP, x represents gene expression and y represents phenotype. We are unable to distinguish causality from pleiotropy. When there are multiple causal variants, the effect size of x on y (b_{xy}) estimated at each causal variant can be various, linkage or multiple causal variants with distinct b_{xy} . It is a complicated scenario, which is not easy to interpret. The proportion of multiple underlying causal variants seems to be smaller than a single causal variant. Using PMR-Egger under the assumption of polygenic model, the causal effect can be considered to be average of b_{xy} at each causal variant. The horizontal effect can be considered to be deviation from the mean effect. In the manuscript, authors noticed that cis-SNPs were in high LD, "GSMR and MR-Egger are not feasible to obtain near independent SNPs". Using GCTA-COJO, what is the proportion of genes that have multiple independent signals?

1.3 The unrealistic assumptions may cause issues in the simulations and real data analysis. For example, 1) Fig 2a showed PMR-Egger had larger power than the other methods only when there was "horizontal pleiotropy". The "horizontal pleiotropy" was not explicitly described in the manuscript. 2) The magnitude of horizontal pleiotropy was small, $1e-4$ to $2e-3$, which can be supposed to be deviation from the average b_{xy} . 3) In the real data analysis, many genes described in the main text were from MHC region, e.g. C2, HLA-F, ZKSCAN4. The LD is complicated in MHC region, which is usually excluded in the analysis. 4) SMR results showed large p-values for HERPUD1 ($p=0.1$), ADAM15 ($p=0.14$) and DNAJC27-AS1 ($p=0.05$), which indicates GWAS p-values at the 3 top cis-SNPs are large. More analysis might be required to investigate whether genes have causal effects on phenotypes. 5) In terms of MAPT, I don't think the horizontal pleiotropy can be concluded from the non-significance of PrediXcan analysis conditional on the flanking gene. Suppose two regression models, a) $y = g_1 + e$ and b) $y | g_2 = g_1 + e$, where $y | g_2$ represents conditioning y on gene 2. I don't think the non-significance of model (b) would indicate the horizontal pleiotropy. Is it because two genes are highly correlated? 6) without any regional figures for real data analysis, I am not convinced the identified genes have causal effects on phenotypes.

2. The method requires the SNPs associated with gene expression. What is the threshold to select SNP instruments?

3. Fig 1 showed p-values for SMR were strongly deflated, and p-values for LDA MR-Egger were largely inflated, which looks weird. P-value for SMR may be deflated, because SE is an approximation. Is the strong deflation because of weak instruments? The LDA MR-Egger is not too different from TWAS. For LDA MR-Egger, $b_{xy} = \text{cov}(g_x, g_y) / h^2_x$, where $\text{cov}(g_x, g_y) = b_{zx} * V^{-1} * b_{zy}$, and $h^2_x = b_{zx} * V^{-1} * b_{zx}$. For TWAS, $rg_{xy} = Z_{twas} / \sqrt{n * h^2_y} = b_{zx} * V^{-1} * b_{zy} / \sqrt{b_{zx} * V^{-1} * b_{zx} * b_{zy} * V^{-1} * b_{zy}} = \text{cov}(g_x, g_y) / \sqrt{h^2_x * h^2_y}$.

4. Table 1 shows PMR-Egger can deal with both individual-level and summary-level data. I am not sure whether "both" means accounting for sample overlap or simply accepting individual-level data and summary-level data as input. The summary-level data can be obtained from GWAS analysis. Therefore, all the methods are able to handle individual-level and summary-level data without accounting for sample overlap.

5. Figure 4, 5 and 6 showed lambda for PMR-EGGER was not too different from SMR or PrediXcan, but the number of genes identified by PMR-Egger was much larger than the other methods. It is confusing to me.

Minor comments:

6. Page 8, "Methods – PMR-Egger overview", I think it might be better to move the first 3 paragraphs to results.
7. Page 9, The authors stated "our model no longer requires the exclusion restriction condition of MR, because of fitted horizontal pleiotropy in the model". The statement is not correct. PMR-Egger follows the assumptions of MR-Egger, e.g. INSIDE. Under INSIDE assumption, the causal effect is mediated by gene expression.
8. Page 9, the font size for the last line is not the same.
9. Page 10, the authors stated "at least one of two assumptions is not testable in practice." I don't think it is correct. For assumption 1), the method requires SNPs strongly associated with gene expression, at least genome-wide significant SNPs.
10. Page 12, " $N(0, PVE_{zx}/556)$ ", "PVE" was not described.
11. Page 24, "increasing horizontal pleiotropy (Supplementary Fig. 2e, f)", where are supplementary figure 2 e and f?

Point-by-point Responses to Reviewer 1

The manuscript describes an approach for identifying a causal effect of gene expression on a complex trait using a likelihood framework called PMR-Egger. The assumptions underlying the likelihood framework is relaxed compared with previous (strict) MR approaches in that it allows for a particular direct effect of SNPs on trait (ie single shared horizontal pleiotropic effect). The approach is sound and unifies various approaches in a single statistical model. The authors demonstrate that PRM-Egger is unbiased under various simulations and apply it to GWAS data from WTCCC, GERA, and UKBB studies. I think the approach is quite nice and the simulations are thorough; however, I do have some comments that should be addressed.

Responses: Thank you for your positive review and constructive comments. Our detailed responses are listed below.

Main comments

1. Across all simulations the authors used a single gene, BACE1, as a basis for comparison. There is some justification for why this gene was selected, but I think it would be much more informative to see the performance of PMR-Egger (and its comparison) across genes selected at random. The reason being that LD patterns vary dramatically and I worry that results are not reflective of how it will be used in real data.

Responses: Thank you for the comments. Following your suggestion, we have performed new cross-gene based simulations. Specifically, we randomly selected 10,000 genes from the total 15,810 genes we analyzed. We extracted cis-SNPs for these genes, obtaining a median of 576 cis-SNPs per gene (min=11; max=7,409). For each gene in turn, we used its cis-SNPs in GEUVADIS to simulate its gene expression level. We also used the same set of cis-SNPs in GERA to generate the GWAS phenotype. Afterwards, we applied different methods to analyze these simulated data. By performing simulations in a cross-gene fashion, the new simulation results will reflect the varying LD pattern (and the varying number of cis-SNPs) across genes that we observe in real data sets. We performed cross-gene simulations under all simulation settings used for the single gene-based simulations, including settings with varying gene expression heritability, varying genetic architectures underlying gene expression, as well as varying causal and horizontal pleiotropy effects. The simulations details are provided in the updated Materials and Methods section (lines 285-294 on page 14 and 15).

The new results from cross-gene based simulations are consistent with (and

almost identical to) the previous simulation results based on a single gene and are also consistent with previous real data results. These new cross-gene simulation results are provided in newly added Supplementary Fig. 7-12, 16-17, 19-20, 24-25, with results details provided in the updated Results section (lines 500-502 on page 24, lines 538-540 on page 26, lines 564-566 on page 27, and lines 588-590 on page 28).

2. It isn't clear to me if other methods are compared using respective individual-level approach or summary-based approaches. This should be much more explicit (both for simulations and real data).

Responses: We apologize for not providing these important details, which are now provided in the updated Materials and Methods section (lines 399-408 on page 19 and 20). Briefly, in all simulations and real data applications, methods that can use either individual-level data or summary statistics (PrediXcan and TWAS) are applied using individual-level data as input to ensure their optimal performance. Methods that can only use individual-level data (CoMM) are applied using individual-level data as input. Methods that can only use summary statistics (SMR and LDA MR-Egger) are applied using summary data as input. For PMR-Egger, we used individual-level data for all main analyses and used summary data only for validating the summary data version of our method described in the Discussion section.

3. Why are results for LDA MR-Egger so extreme? LDA MR Egger is essentially TWAS but with an added intercept, which should reduce overall power. Based on my reading it also estimates an additional variance term compared with standard TWAS, which should further reduce its power in comparison. I understand that the authors did not design MR Egger, but the results seem more like a bug.

Responses: Thank you for the comments. We had the exact same concern when we first saw our results from LDA MR-Egger a few years ago. We initially also thought it was a software coding bug, especially considering that LDA MR-Egger software was not publicly available when we started the project at that time and we had to implement our own version of LDA MR-Egger at the beginning. Therefore, we reached out to the first and last authors of the LDA MR-Egger paper last year. These authors kindly provided us with their code, which was used to produce all the results in the present study. We found that the results from authors' code are identical to those obtained with our earlier implementation. In addition, we were also able to replicate all results in the LDA MR-Egger paper using either their code or our code. We further carefully investigated their model in detail and found out one key issue that makes LDA MR-Egger unsuitable for TWAS settings; that is, LDA MR-Egger treats the SNP

effects on gene (the parameters β in our paper) as fixed effect. (Note that there is no variance component term in the LDA MR-Egger model. The “an additional variance term” mentioned in your comment perhaps is the standard error for the estimate of the marginal SNP on gene effects, which is denoted as Σ_G in the LDA MR-Egger paper.) However, fixed effect size is not a good assumption for TWAS applications as the number of cis-SNPs is often on the same order as the number of individuals in the gene expression study. The fixed effect assumption, when paired with the two-stage inference procedure that ignores the estimation uncertainty in the first stage, makes LDA MR-Egger sensitive to the collinearity induced by SNP correlations caused by LD.

Indeed, we found that the LDA MR-Egger results are ok if we follow the same simulation setting used in the LDA MR-Egger paper, where SNP genotypes are simulated based on an autoregressive covariance matrix with a relatively moderate correlation parameter. However, when such correlation parameter is set to be realistically high (>0.9) or if we use SNPs from real data to carry out the same set of simulations as described in the LDA MR-Egger paper, then the results from LDA MR-Egger software become extreme. Importantly, we can also recapitulate the extreme behavior of LDA MR-Egger in the real data applications as well as the new cross-gene based simulations described in the response to your first comment, suggesting that the LD pattern of *BACE1* gene used in the previous single-gene based simulations is reasonably representative of LD structure across genes and is not the cause of extreme behavior of LDA MR-Egger.

We showed these results described in the above paragraph in the Supplementary Fig. S2 in the previous version of manuscript. We also provided a short explanation of the extreme LDA MR-Egger behavior in the previous Results section. Based on the reviewer’s comments, we realized that the short explanation in the previous version of manuscript was not sufficient. Therefore, in the updated manuscript, we provided a more elaborated explanation in the Results section in line of what explained above (lines 456-468 on page 22).

4. The authors perform nice simulations demonstrating unbiased estimates of the pleiotropic effect under various perturbations to model assumptions. Can the authors perform a set of simulations that randomly flips the encoding at a subset of genotypes prior to the estimation procedure. It is not clear how the approach can know a-priori what the correct encoding is to estimate the directional effect over all data.

Responses: Thank you and we apologize for a potential miscommunication here. Previously, we performed simulations where we flipped the sign of the pleiotropic effect for a fixed proportion of SNPs; the proportions were set to be 10%, 30% or 50%

to create three simulation settings. These three simulation settings include two approximately directional pleiotropy settings where the ratio of SNPs with negative vs positive effects is 1:9 (i.e. flip 10%) or 3:7 (i.e. flip 30%); and one balanced setting where the ratio of SNPs with negative vs positive effects is 5:5 (i.e. flip 50%). These flip sign simulations are effectively what you suggested above as flipping the genotype encoding.

In these flip sign simulations, we found that the type I error for testing the *causal effect* remains calibrated in either the approximately directional pleiotropy settings or the balanced setting when horizontal pleiotropic effect is small or moderate ($\gamma = 1 \times 10^{-4}$, 5×10^{-4} , or 1×10^{-3} ; Supplementary Fig. 6a, b, c). However, when horizontal pleiotropic effect is large ($\gamma = 2 \times 10^{-3}$), as one would expect, the p-values from PMR-Egger becomes inflated, with the genomic control factor being 1.08, 1.31 and 1.37, for settings where the ratio is 1:9, 3:7 and 5:5, respectively (Supplementary Fig. 6d). We did not previously examine estimates for the *pleiotropic effect* in the flip sign simulations since a single scalar γ employed in our model is no longer expected to capture the complex pleiotropic effects. Thus, we previously only showed that, in the presence of directional pleiotropic effect (i.e. absence of sign flipping), PMR-Egger can estimate the horizontal pleiotropic effect size accurately (Supplementary Fig. 23).

Following your comment, we performed additional analysis to examine in detail the consequences of effect size flipping (newly added Supplementary Fig. 15). We found that (1) the *causal effect* estimates remain reasonably unbiased in both the approximately directional pleiotropy settings and the balanced setting (results added to lines 536-538 on page 25 and 26); (2) as expected, in the absence of directional pleiotropy, the estimates of *pleiotropic effects* are under-ward biased, more so in the balanced setting than in the approximately directional pleiotropy settings (results added to lines 586-588 on page 28); (3) the power for detecting *pleiotropic effects* in the hypothesis test also reduces in the absence of directional pleiotropy (results added to lines 582-583 on page 27). We have added these new results to the Results section.

We also fully agree with the reviewer that we do not know *a priori* whether the directional pleiotropy assumption is correct or not. We have previously attempted to alleviate this restrictive modeling assumption by imposing an alternative modeling assumption on the horizontal effect sizes based on variance component assumption (which we termed as PMR variance component model in the Discussion). Unfortunately, we spent more than half a year on the PMR variance component model but couldn't obtain calibrated p-values for causal effect testing at the genome-wide threshold for TWAS applications. We previously mentioned this important drawback of PMR-Egger in the second paragraph of Discussion. Following your comment, we modified that second paragraph to emphasize this important issue further (lines 814-845 on pages 38-39).

5. In real data, authors should include genotyping PCs to minimize the impact of ancestry/stratification. While this doesn't fit the model as explicitly defined (single μ_x) it should be estimable and adjusted for under a fixed-effect model ($\mu_x = X\beta$) as is standard in other approaches.

Responses: Thank you for the comments. In the previous manuscript, we failed to mention that we regressed phenotypes on the top 10 genotyping PCs to obtain the phenotype residuals, which we used further to conduct TWAS analysis with different methods. Following your suggestion, we have added these important details in the Materials and Methods section (lines 324-330 on page 16; lines 345-350 on page 17; lines 364-366 on page 18). In addition, following your suggestion, we have performed additional analysis where we used the original phenotypes and included the top 10 genotyping PCs as covariates in the model, in parallel to the previous analysis using phenotype residuals. The new results are largely consistent with previous results. The new results with PCs added as covariates are also provided in the updated Supplementary Fig. 29-31 and briefly described in the updated Results section (lines 604-608 on pages 28-29).

6. Why does TWAS have such a large average (and std) time for its first entry in Table 1? Is this the result of a single outlier?

Responses: Thank you for spotting the error (due to typo), which we have corrected in the revised Table 1 (the correct result is 1.97+/-0.86).

7. Genomic control estimates reflect differential power in WTCCC/GERA vs UKBB studies

Responses: Indeed, the genomic control estimates reflect at least in part the different power in WTCCC/GERA vs UKBB studies under polygenic genetic architectures. We have mentioned this point in the revised manuscript (lines 621-624 on page 29).

8. The manuscript describes several examples where associations go away after controlling for nearby genes as evidence in support of pleiotropic effects. An alternative and simpler explanation for these examples is that SNPs used in the model are simply tagging nearby eQTLs (or some molecular feature) rather than directly impacting downstream trait through other pathways. This approach is modeled in a recent method FOCUS.

Responses: Thank you and we fully agree. In the updated Results section, we

acknowledge that these listed examples are all focused on the special case where the false gene association with the trait disappears when conditional on a neighboring gene. It is not straightforward to provide general examples where the apparently false gene association with the trait may be explained by horizontal pleiotropic effects acted upon a gene far away, as it is extremely challenging to convincingly identify trans eQTL effects. Subsequently, we fully acknowledge that, in these examples we focused on, while it is possible SNPs display true horizontal pleiotropic effects through the neighboring gene, it is equally or more likely that SNPs used in the model are simply tagging nearby eQTLs of the causal gene (as employed in the approach of FOCUS¹) and thus display apparent “horizontal pleiotropic effect” through the neighboring gene. Subsequently, the horizontal pleiotropic effect term in PMR-Egger may represent the apparent “horizontal pleiotropic effect” through SNP tagging to the nearby eQTLs of the causal gene, rather than the truly pleiotropic effect acted through other molecular pathways. Regardless of the interpretation of the pleiotropic effect term, however, we found it reassuring that by modeling the pleiotropic effect term in PMR-Egger can reduce false discoveries in the case of SNP tagging. We have updated the Results section to mention this important point and cited the FOCUS paper there (lines 771-784 on page 36). In addition, we have added FOCUS into the real data applications (more details in the response to your next comment below).

9. FOCUS is a recently published tool that attempts to control for pleiotropic effects in TWAS by jointly modelling multiple genes at a GWAS region. Can the authors compare, at a minimum, real-data results of PMR-Egger with FOCUS?

Responses: Thank you. Following your suggestion, we applied the recently published method FOCUS to all real data applications. During this analysis, we realized that it is not straightforward to directly compare the results of PMR-Egger with that of FOCUS, as the two are focused on two completely different tasks and thus use different analytic procedures. Specifically, PMR-Egger and the other TWAS/MR methods are directly applied to analyze all genes. In contrast, FOCUS is a fine mapping method that is used to analyze a small subset of genomic regions that contain at least one candidate gene and at least one significant SNP. As a result, FOCUS is often used to analyze a much smaller set of genes as compared to the other TWAS/MR methods. Besides this important difference, FOCUS is also a Bayesian approach that outputs a posterior inclusion probability for each gene instead of a p-value as its association evidence. Subsequently, it is not straightforward to compare PMR-Egger with FOCUS directly in terms of type I error control and power. However, through fine mapping, the identified genes in the credible set output by FOCUS likely represent the truly causal genes. Therefore, we can treat the FOCUS output results as the ground truth and compare different methods with FOCUS to infer their power in the real data

applications.

In each real data set, following the original FOCUS paper and the default setting of FOCUS, we focused on regions that harbor at least one genome-wide-significant SNP ($p < 5 \times 10^{-8}$), and for each TWAS/MR method (i.e. PMR-Egger, TWAS, PrediXcan, CoMM, or SMR), also harbor at least one TWAS gene that is declared significant by the given method. We analyzed a total of 653 genes in 30 regions in WTCCC, 892 genes in 47 regions in GERA, and 35345 genes in 1441 regions in UK Biobank. We detected a total of 15, 35, and 2083 genes in the 90% credible set by FOCUS in WTCCC, GERA, and UK Biobank, respectively. Due to the small number of genes detected in the credible set in WTCCC, we focus our main comparison in the GERA and UK Biobank data. The FOCUS analysis details are provided in the Materials and Methods section (lines 409-417 on page 20).

In these real data applications, we found that the results from PMR-Egger is largely consistent with that of FOCUS, more so than the other TWAS/MR methods. Specifically, the average PMR-Egger $-\log_{10}(\text{p-value})$ for genes in the FOCUS 90% credible set is 22.43 in GERA and 10.67 in UK Biobank. The average $-\log_{10}(\text{p-value})$ of PMR-Egger is higher than CoMM (13.83 and 10.43), TWAS (5.71 and 7.55), PrediXcan (4.66 and 7.06) and SMR (NA for GERA, as no gene in the credible set is detected by SMR; 1.78 for UK Biobank). In addition, the difference of the average PMR-Egger $-\log_{10}(\text{p-value})$ between genes in the FOCUS credible set and genes outside is large (16.61 in GERA and 7.43 in UK Biobank). The $-\log_{10}(\text{p-value})$ difference is again larger than CoMM (8.41 and 6.02), TWAS (4.52 and 5.35), PrediXcan (3.50 and 4.74) and SMR (NA and 0.28). Similarly, the proportion of significant genes detected by PMR-Egger in the FOCUS credible set is 78% in GERA and 60% in UK Biobank. The proportion of significant genes by PMR-Egger is higher than CoMM (75% and 53%), TWAS (50% and 47%), PrediXcan (50% and 48%) and SMR (NA and 8%). In addition, the difference in the proportion of significant genes detected by PMR-Egger between genes in the FOCUS credible set and genes outside is high (53% in GERA and 41% in UK Biobank). This proportion difference by PMR-Egger is again higher than CoMM (51% and 39%), TWAS (46% and 36%), PrediXcan (50% and 35%) and SMR (NA and 1%). The new results are shown in Supplementary Table 3 and Supplementary Figure 41, with details provided in the Results section (lines 698-723 on page 33 and 34).

10. The authors describe a very nice extension of their approach to estimate the causal (and pleiotropic) effect from summary data. Considering the summary based approach will likely be the most common use-case scenario it is important that the authors demonstrate its performance compared with the individual-level approach.

Responses: Thank you for the comment. Following your suggestion, we validated our implementation of the summary statistics-based approach in simulations. Simulation details are provided in the updated Materials and Methods section (lines 274-284 on pages 14). Briefly, in the comparison, we constructed the SNP by SNP correlation matrix from three different reference panels: all individuals from the GWAS data; 10% randomly selected individuals from the GWAS data; individuals of European ancestry from the 1,000 Genomes project. We then applied the summary statistics based approach of PMR-Egger to each reference panel and compared results with the individual level data based approach of PMR-Egger that was applied to the complete data. As expected, we found that the p-values for testing the causal effects and the p-values for testing the pleiotropy effects are consistent between the summary statistics based approach and the individual data based approach. These new results demonstrate the correct implementation and effectiveness of the summary statistics based approach of PMR-Egger. The results are provided in the new Supplementary Fig. 43, with details provided in the Discussion section (lines 863-874 on pages 40-41). The summary statistics based approach of PMR-Egger is also implemented in the open source software package along with the individual level data based approach.

Minor comments

1. I would like to commend the authors for including their source code along with the manuscript.

Responses: Thank you. In addition to including the PMR-Egger source code along with the manuscript, we have also posted the software on R CRAN for easy installation and usage (<https://cran.r-project.org/web/packages/PPMR/index.html>). We will also keep updating the software in the previously provided github link.

Reference

- 1 Mancuso, N. *et al.* Probabilistic fine-mapping of transcriptome-wide association studies. *Nature genetics* **51**, 675 (2019).
- 2 Berisa, T. & Pickrell, J. K. Approximately independent linkage disequilibrium blocks in human populations. *Bioinformatics* **32**, 283 (2016).
- 3 Zhou, X., Carbonetto, P. & Stephens, M. Polygenic Modeling with Bayesian Sparse Linear Mixed Models. *Plos Genetics* **9**, : e1003264. (2013).
- 4 Gusev, A. *et al.* Integrative approaches for large-scale transcriptome-wide association studies. *Nature genetics* **48**, 245-252 (2016).
- 5 Gusev, A. *et al.* Transcriptome-wide association study of schizophrenia and chromatin activity yields mechanistic disease insights. *Nature genetics* **50**, 538 (2018).

Point-by-point Responses to Reviewer 2

Thank you for the opportunity to review this manuscript. The authors have put together a method with the aim of distinguishing causality from horizontal pleiotropy in transcriptome-wide association studies. Whilst this is an area in the field where methods development is needed, I'm afraid that I have several concerns regarding this approach. This in particular is highlighted by the real data examples chosen in the study, amongst which the authors suggest their method implicates HERPUD1 as the causal gene responsible for an association signal with dyslipidemia on chromosome 16. This gene neighbors CETP, perhaps the most well-known of lipid gene to date. I have attached my more detailed comments below in case the authors find them useful in refining their manuscript.

Responses: Thanks for your positive review and constructive comments. Our detailed responses are provided below.

Page 6 – The authors suggest that Cochran's Q can test for horizontal pleiotropy, although this is misleading. This test is commonly used to assess in 2-sample MR analyses, however the presence of heterogeneity does not mean that a causal relationship does not underly an association. I recommend that this should be removed.

Responses: Thank you and we apologize for the error. We have removed that part of the sentence on Cochran's Q (line 107-109 on page 6).

Page 8 – At the start of the methods section the authors refer to gene expression as an 'explanatory variable' in their analysis. I assume they mean it's being treated as their 'exposure' in an MR setting?

Responses: Yes, indeed. For consistency, we have replaced "explanatory variable" with "exposure" throughout the text.

Page 10 – My main concern with this method is that it relies on an assumption used in rare variant analyses (i.e. the burden test method) that variants are independent. The authors state that under this assumption their model 'reduces to the commonly used MR-Egger regression model', which of course relies of the assumption of independent instruments. Moreover, in a conventional MR setting the MR-Egger approach only becomes powerful in terms of distinguishing horizontal pleiotropy when many (i.e. dozens) of independent genome-wide

(strong) instruments are used. This is of course incredibly unrealistic for cis-eQTL, particularly using a sample size of n=465.

Response: We apologize for this miscommunication and for creating a false impression that our method only handles independent SNPs. In contrast, a key innovation of our method is its use of correlated SNPs and its ability to account for SNP correlation. Because PMR-Egger relies on a likelihood framework to account for SNP correlation, our method is able to extend the commonly applied MR-Egger to TWAS settings where SNPs are all highly correlated with each other. Through the likelihood framework, PMR-Egger also unifies many existing TWAS and MR methods into the same probabilistic modeling framework. We previously used a large proportion of the 3rd paragraph in the Introduction section to emphasize the importance of using correlated SNPs for TWAS applications. We previously listed this key modeling innovation in Table 1 along with many existing MR and TWAS methods. We previously described the use of correlated SNPs in the Methods section (method overview, simulations, and real data applications subsections). All our previous simulations and real data applications used all cis-SNPs that are correlated and in LD with each other. We also tried to convey this key message multiple times through Abstract, Results and Discussion sections.

Following your comment, we realize that our previous emphasis was insufficient. We re-examined our text and we suspected this key miscommunication stems from a long sentence in the Methods section: *“With the burden test assumption on γ , in the special case where instruments are independent and treated as fixed effects and where a two-stage estimation procedure is used for inference, our model reduces to the commonly used MR-Egger regression model.”*. When the middle part of the sentence, *“in the special case where instruments are independent and treated as fixed effects and where a two-stage estimation procedure is used for inference”*, is ignored, then this sentence does give the false impression that PMR-Egger is a simple probabilistic extension of MR-Egger and that PMR-Egger can only handle independent SNPs like MR-Egger does. The middle part, specifically *“in the special case where instruments are independent and treated as fixed effects”*, was trying to emphasize that only in these special cases PMR-Egger would reduce to MR-Egger; but PMR-Egger can accommodate more general cases and effectively extends MR-Egger to accommodate correlated instruments. To avoid miscommunication, we have rewritten that sentence into four short sentences: *“With the burden test assumption on γ , our model becomes a generalization of the commonly used MR-Egger regression model. In the special case where instruments are independent and treated as fixed effects and where a two-stage estimation procedure is used for inference, our model reduces to MR-Egger. However, our method can handle general cases where MR-Egger does not apply to. In particular, unlike MR-Egger, our method can handle multiple correlated*

instruments and perform inference in a likelihood framework.” (lines 197-203 on pages 10-11). In addition, we have added message related to correlated SNPs to multiple places throughout the text to emphasize the importance of modeling correlated SNPs. For example, we now explicitly mention that we used all cis-SNPs for PMR-Egger in all our analyses (lines 149-151 on page 8; lines 399-400 on page 19).

I recommend that the authors test this implicitly in their paper i.e. simulating a correlation structure amongst eQTL and assessing whether false positive rates increase in the presence of higher correlation. If it does then perhaps incorporating a reference panel, similar to conventional 2-sample MR analyses, may assist in removing tightly correlated instruments that may bias findings.

Responses: Thank you. Indeed, as explained in the response to your previous comment, all our previous simulations and real data applications all cis-SNPs – these SNPs are all highly correlated with each other due to LD. Following TWAS tradition, we did not perform any pruning for most TWAS methods including PMR-Egger/TWAS/PrediXcan/CoMM. In the simulations, the eQTLs were always simulated at random; thus, in the simulation setting where there are multiple eQTLs, these eQTLs were also correlated with each other due to LD. In the summary statistics version of our method, we also used a reference panel to compute SNP correlations. To avoid future confusion, we have further emphasized this important point of modeling correlated SNPs throughout the text (details in the response to your previous comments). For example, we now explicitly mention that we used all cis-SNPs for PMR-Egger in all our analyses (lines 149-151 on page 8; lines 399-400 on page 19).

Page 26 – In the real data application of their method, the authors state that ‘Overall, by controlling for horizontal pleiotropic effects, PMR-Egger detected many likely causal genes that the other methods failed to detect.’ Conventionally these sections of papers concern cherry-picked examples where their method outperforms competitors. However, I find the examples quite concerning. The first 2 examples are both in the HLA region of the genome (C2 & HLA-F), which is a region of the genome many previous approaches do not attempt to investigate due to the extensive LD structure here making disentangling causal genes extremely problematic. I am therefore very skeptical that these genes are likely the causal genes for association signals in this region, particularly giving concerns regarding independent eQTL not being accounted for. For example, did the authors assess whether eQTL for these genes in their dataset are also eQTL for various neighboring genes? My guess is that there would be a lot of genes where this is the case. I would strongly recommend choosing genes outside of

HLA to showcase your method. Even more alarming is the third example – which is HERPUD1 and its association with dyslipidemia. The authors state this is a ‘well-known lipid-related gene’ and cite 2 genetic association studies. I was unable however to find this gene symbol in either study. It may be tucked away in the supplementary material somewhere. However, I stopped looking when discovering that this gene neighbors CETP, which is actually a well-known lipids gene. The association signal at the locus is therefore most likely due to CETP, and HERPUD1 I imagine is simply co-expressed with it. I would again strongly encourage the authors to take this example out of their manuscript and consider alternatives. Again, across ~20,000 coding genes some cherry-picking to best showcase a method is not uncommon in these types of studies. However, this makes the choice of these 3 examples slightly worrying.

Responses: Thank you. We really appreciate this comment and we apologize for not being careful in selecting these previous examples. Following your suggestion, we have re-run all our real data analysis by excluding the MHC region and have removed all the three examples. We have listed three new gene examples that have convincing previous biological literature support on their causality. These three examples include:

1. The *LNK/SH2B3* gene (111,743,752-111,989,427 on chr 12) is only identified by PMR-Egger to be associated with platelet count in the UK Biobank (PMR-Egger $p = 1.17 \times 10^{-221}$; CoMM $p=0.98$; TWAS $p = 8.6 \times 10^{-5}$; PrediXcan $p=0.68$; SMR $p=0.024$). The association between *LNK* and plate count is consistent with results from recent large-scale GWASs¹⁻³. *LNK/SH2B3* encodes the lymphocyte adaptor protein (LNK) that is primarily expressed in hematopoietic and endothelial cells⁴. In hematopoietic cells, LNK functions as a negative regulator of cell proliferation as well as the thrombopoietin-mediated cytokine signaling pathway, which is a key signaling pathway that promotes megakaryocytes to form platelets^{4,5}. Indeed, platelets are overproduced and accumulated in *LNK* knockdown cells as well as *Lnk* knockout mouse⁶⁻⁸, supporting a causal role of *LNK* in platelets production.

2. The *NOD2* gene (50,627,514-50,866,988 on chr 16) is identified by PMR-Egger to be associated with Crohn’s disease (CD; $p = 6.1 \times 10^{-19}$), and, with a slightly less significance, also by CoMM ($p = 7.8 \times 10^{-15}$). The association between *NOD2* and CD was not identified by the other methods (TWAS $p=0.005$; PrediXcan $p=0.92$; SMR $p=0.15$). *NOD2* encodes a cytosolic pattern recognition receptor that acts both as a cytoplasmic sensor of microbial products and as an important mediator of innate immunity and inflammatory response⁹ The *NOD2* gene is a well-known susceptible gene for CD and is perhaps one of the first genes ever implied for CD¹⁰. Multiple SNPs in *NOD2* have been found to be associated with CD in both early linkage studies¹¹⁻¹³ and many recent GWASs^{14,15}. *NOD2* variants associated with CD often reside in the ligand recognition domain of *NOD2* and can lead to aberrant bacterial

handling and antigen presentation¹⁶. Indeed, *NOD2*-deficient mice displays dysregulated bacterial community in the ileum and *NOD2*-deficient ileal epithelia exhibit impaired ability of inducing immune responses for bacteria elimination¹⁷. It is thus hypothesized that mis-regulation of *NOD2* can causally lead to altered interactions between ileal microbiota and mucosal immunity, resulting in increased disease susceptibility to CD¹⁷.

3. The *TFRC* gene (195,654,054-195,909,060 on chr 3) is identified by PMR-Egger to be associated with red blood cell distribution width (RDW) in the UK Biobank ($p = 3.3 \times 10^{-17}$). Such association is not identified by the other methods (CoMM $p=0.95$; TWAS $p=0.76$; PrediXcan $p=0.97$; SMR $p=0.38$). *TFRC* encodes the classical transferrin receptor that is involved in cellular iron uptake^{18,19}. Multiple SNPs in *TFRC* have been established to be associated with various erythrocyte phenotypes in GWASs^{20,21}. These associated erythrocyte phenotypes include the mean corpuscular hemoglobin (MCH) and mean corpuscular volume (MCV, the average volume of red blood cells) which is directly related to RDW^{19,20}. The variants in *TFRC* likely lead to decreased iron availability for red cell precursors, as has been observed in mice deficient in *TFRC*, thus resulting in a compensatory increase of red blood cell size as measured by RDW²².

Details of these are updated in the Results section (lines 661-697 on page 31-32). In addition, the regional association plots (i.e. locus zoom plots) for these three genes are also presented (the newly added supplementary Fig.33-35).

Minor comments:

Page 5 – ‘multiple independent/near-independent SNP instruments’ – strictly instrumental variables for established 2-sample MR approaches should be independent – it’s only how one defines ‘independent’ that allows ‘near-independent’ SNPs to creep into analyses

Responses: Thank you for the comment. We completely agree with the reviewer that the use of “near-independent” is not scientifically rigor. We used this terminology only because this is what was used in the original GSMR paper. Following the comment, we have deleted “*near-independent*” throughout the text and Table 1.

Page 5 – minor typo ‘suggest’ should be ‘suggests’ on last line

Responses: Thank you for pointing out this error, which we have corrected in the revised version.

Page 6 – ‘Cohran’s’ should be ‘Cochran’s’

Responses: Thank you for pointing out this error, which we have corrected in the

revised version.

Some of the grammar and language is slightly hard to follow throughout.

Responses: Thank you. We have made minor text modifications throughout the paper to make it easier to follow.

Reference

- 1 Kamatani, Y. *et al.* Genome-wide association study of hematological and biochemical traits in a Japanese population. *Nature genetics* **42**, 210 (2010).
- 2 Soranzo, N. *et al.* A genome-wide meta-analysis identifies 22 loci associated with eight hematological parameters in the HaemGen consortium. *Nature genetics* **41**, 1182 (2009).
- 3 Auer, P. L. *et al.* Rare and low-frequency coding variants in CXCR2 and other genes are associated with hematological traits. *Nature genetics* **46**, 629 (2014).
- 4 Bersenev, A., Wu, C., Balcerak, J. & Tong, W. Lnk controls mouse hematopoietic stem cell self-renewal and quiescence through direct interactions with JAK2. *The Journal of clinical investigation* **118**, 2832-2844 (2008).
- 5 Tong, W. & Lodish, H. F. Lnk inhibits Tpo-mpl signaling and Tpo-mediated megakaryocytopoiesis. *Journal of Experimental Medicine* **200**, 569-580 (2004).
- 6 Takizawa, H. *et al.* Lnk regulates integrin $\alpha\text{IIb}\beta\text{3}$ outside-in signaling in mouse platelets, leading to stabilization of thrombus development in vivo. *The Journal of clinical investigation* **120**, 179-190 (2010).
- 7 Viny, A. D. & Levine, R. L. Genetics of myeloproliferative neoplasms. *Cancer journal (Sudbury, Mass.)* **20**, 61 (2014).
- 8 Bersenev, A. *et al.* Lnk constrains myeloproliferative diseases in mice. *The Journal of clinical investigation* **120**, 2058-2069 (2010).
- 9 Yamamoto, S. & Ma, X. Role of Nod2 in the development of Crohn's disease. *Microbes and infection* **11**, 912-918 (2009).
- 10 McGovern, D., Van Heel, D., Ahmad, T. & Jewell, D. NOD2 (CARD15), the first susceptibility gene for Crohn's disease. *Gut* **49**, 752-754 (2001).
- 11 Hugot, J.-P. *et al.* Mapping of a susceptibility locus for Crohn's disease on chromosome 16. *Nature* **379**, 821 (1996).
- 12 Hugot, J.-P. *et al.* Association of NOD2 leucine-rich repeat variants with susceptibility to Crohn's disease. *Nature* **411**, 599 (2001).
- 13 Ogura, Y. *et al.* A frameshift mutation in NOD2 associated with susceptibility to Crohn's disease. *Nature* **411**, 603 (2001).
- 14 Franke, A. *et al.* Genome-wide meta-analysis increases to 71 the number of confirmed Crohn's disease susceptibility loci. *Nature genetics* **42**, 1118 (2010).
- 15 Franke, A. *et al.* Replication of signals from recent studies of Crohn's disease identifies previously unknown disease loci for ulcerative colitis. *Nature genetics* **40**, 713 (2008).
- 16 Kennedy, N. A. *et al.* The impact of NOD2 variants on fecal microbiota in Crohn's disease and controls without gastrointestinal disease. *Inflammatory bowel diseases* **24**, 583-592 (2018).
- 17 Sidiq, T., Yoshihama, S., Downs, I. & Kobayashi, K. S. Nod2: a critical regulator of ileal microbiota and Crohn's disease. *Front Immunol* **7**, 367 (2016).

- 18 Keel, S. B. *et al.* Evidence that the expression of transferrin receptor 1 on erythroid marrow cells mediates hepcidin suppression in the liver. *Experimental hematology* **43**, 469-478. e466 (2015).
- 19 Andrews, N. C. Genes determining blood cell traits. *Nature genetics* **41**, 1161 (2009).
- 20 Ganesh, S. K. *et al.* Multiple loci influence erythrocyte phenotypes in the CHARGE Consortium. *Nature genetics* **41**, 1191 (2009).
- 21 Lo, K. S. *et al.* Genetic association analysis highlights new loci that modulate hematological trait variation in Caucasians and African Americans. *Human genetics* **129**, 307-317 (2011).
- 22 Levy, J. E., Jin, O., Fujiwara, Y., Kuo, F. & Andrews, N. Transferrin receptor is necessary for development of erythrocytes and the nervous system. *Nature genetics* **21**, 396 (1999).

Point-by-point Responses to Reviewer 3

The authors presented the developed method PMR-Egger, estimating both causal and pleiotropic effects simultaneously. PMR-Egger follows the framework and the assumptions of MR-Egger, but using likelihood test to do the tests and estimate the effects. The simulations showed PMR-Egger was well calibrated and had larger power than existing methods. The real data analysis showed more genes were detected by PMR-Egger. However, the underlying biological model is not clear to me. Therefore, I am not convinced about the results.

Responses: Thank you for your constructive comments. Our detailed responses are provided below. Reading through your comments, we realized that our previous manuscript must have failed to provide sufficient introduction and background for standard TWAS applications, which lead to reviewer's confusion of our "**underlying biological model**", resulting in the reviewer "**not convinced about the results**". In fact, as we will explain in more details to your comment 1, our method makes the standard polygenic modeling assumptions that are directly in line with the polygenic modeling assumptions made in almost all previous TWAS methods (TWAS¹, PrediXcan², CoMM³, DPR⁴, TIGAR⁵ etc.). We sincerely apologize for this unfortunate miscommunication we made in the previous manuscript that leads to reviewer's main concern. We hope the detailed responses to your comments and the updated text can now provide more clarity on this important issue.

Major comments

1. The underlying biological model is unclear. From my understanding, there are not many causal variants that have effects on gene expressions. The PMR-Egger analyses were done under the assumption of polygenic model, which is opposite to what I thought. Moreover, they assumed many cis-SNPs may have pleiotropic effects, which I do not understand.

Responses: We apologize for not providing enough background for TWAS analysis in the previous manuscript, which leads to this misunderstanding. Contrast to what the reviewer thought, almost all previous TWAS methods (TWAS¹, PrediXcan², CoMM, DPR⁴, TIGAR⁵ etc.) make a polygenic modeling assumption and assume that cis-SNPs have non-zero polygenic effects on gene expression. Specifically, TWAS makes the BSLMM polygenic modeling assumption: all cis-SNPs have non-zero effects and their effect sizes follow a mixture of two normal distributions. PrediXcan makes the ElasticNet modeling assumption: all cis-SNPs have non-zero effects *a priori* and their effect sizes follows a mixture of Laplace (L1) and normal (L2) distributions. Both TIGAR and DPR makes the Bayesian non-parametric polygenic

modeling assumption: all cis-SNPs have non-zero effects and their effect sizes follow a mixture of many normal distributions. CoMM makes the standard polygenic modeling assumption: all cis-SNPs have non-zero effects and their effect sizes follow a normal distribution. Perhaps the only method previously used in TWAS setting that makes a sparse modeling assumption is SMR. However, as has been demonstrated in the previous literature (e.g. S-PrediXcan⁶ and here, SMR is not as powerfully as the methods that make polygenic modeling assumptions for TWAS applications. The modeling assumptions of all these methods were previously listed in the Table 1. Following your comment, we have also added a short sentence in Introduction (lines 84-90 on page 5) and created a short literature survey in the Supplementary Text (lines 214-233 on page 8-9) to introduce this background of previous TWAS methods.

The polygenic modeling assumption made in most existing TWAS methods are consistent with the previously known fact that polygenic models often outperform sparse models in gene expression prediction^{4,7} and are also consistent with our current results showing that polygenic models also outperform sparse models for TWAS applications. Therefore, it is important to make polygenic/omnigenic modeling assumptions on β for TWAS applications.

We suspect the reviewer's misconception originates from two places. **First**, while the modeling assumption underlying PrediXcan is polygenic, it does use an optimization algorithm that obtains the posterior mode estimates (which is sparse) instead of the posterior mean estimates (which is polygenic). However, it is important to distinguish the modeling assumption from the inference algorithm. In addition, as we explain more below to your comment 1.3.1, using the sparse posterior mode estimates does make PrediXcan less powerful as compared to the other TWAS methods that obtain polygenic estimates (e.g. PMR-Egger/TWAS/CoMM).

Second, the current eQTL mapping studies all have small sample sizes and thus have limited power in identifying all eQTLs. Subsequently, the small number of eQTLs identified per gene thus far may have created a false impression that the genetic architecture underlying gene expression must be sparse. However, as we will explain more below to your comment 1.2.2, being only able to identify a small number of independent cis-eQTLs does not imply that there is only a small number of causal SNPs underlying gene expression. Indeed, the currently identified top cis-SNPs can only explain a small proportion of cis-heritability, suggesting that many more eQTLs remain to be discovered in the future larger eQTL mapping studies. Therefore, the genetic architecture underlying gene expression may be highly polygenic or even omnigenic.

In any case, we would like to quote George Box's famous saying: "all models are wrong, but some are useful". Regardless whether the polygenic modeling assumption is "correct" or not, we hope the reviewer at least agree that the polygenic modeling assumption made in most TWAS methods is a useful modeling assumption for TWAS

applications.

1.1 We assume there are few causal variants, i.e. only a few top SNPs can capture most of the genetic variation. SMR assumes there is a single causal variant, selecting the top cis-SNP. In practice, the top cis-SNP may not capture all the genetic variation. Therefore, other methods, eg., “TWAS” or “PrediXcan” has larger power. I don’t think using all the cis-SNPs may gain much extra power. Even in the simulation where many causal variants were simulated, PMR-Egger does not show larger power under the causality model (Fig 2a).

Responses: This comment appears to contain two independent messages that the reviewer wants to convey. We answer these two messages separately below:

First, with regard to **“In practice, the top cis-SNP may not capture all the genetic variation. Therefore, other methods, eg., “TWAS” or “PrediXcan” has larger power. I don’t think using all the cis-SNPs may gain much extra power.”**, we apologize for not providing enough background on TWAS and thus misleading the reviewer into this incorrect statement. As we explained in the response to your early comment, most TWAS methods (TWAS/CoMM/PMR-Egger/DPR/TIGAR) model all cis-SNPs jointly and estimate the SNP effect size for every single SNP; PrediXcan also models all cis-SNPs jointly but assign non-zero effects to a small set of SNPs (as it obtains the posterior mode estimates instead of the posterior mean estimates); only SMR selects the top cis-SNP for analysis. Therefore, the fact that PMR-Egger/CoMM/TWAS performs better than SMR (and PrediXcan) highlights the importance of using all cis-SNPs.

Second, with regard to **“I don’t think using all the cis-SNPs may gain much extra power”**, as **“even in the simulation where many causal variants were simulated, PMR-Egger does not show larger power under the causality model (Fig 2a).”**, we are unsure which part of the figure 2a misleads the reviewer into this incorrect statement. Fig 2a clearly shows that PMR-Egger also outperforms SMR (and PrediXcan), highlighting the importance of modeling all cis-SNPs.

We are wondering whether the reviewer’s second point is referring to a separate issue of comparing PMR-Egger vs TWAS/CoMM, as also mentioned in your comment 1.3.1 below. As we explained previously in the manuscript, the slightly lower power of PMR-Egger as compared to TWAS/CoMM in the absence of horizontal pleiotropy is expected: this is not related to our polygenic modeling assumption on β (since all these three methods make a polygenic modeling assumption), but is simply due to the loss of degree of freedom inevitably occurs when we introduce parameters to model the horizontal pleiotropic term. It is well recognized in the statistical literature that any modeling assumption can lead to a small power loss when the modeling assumption is not satisfied – after all, there is no

free lunch with any statistical models. Because of the additional term to model horizontal pleiotropy, PMR-Egger will lose degrees of freedom in the absence of horizontal pleiotropy. Following your comment, we have now also highlighted this explanation in red to make it apparent to the reviewer (lines 506-512 on page 24).

1.2.1 From my understanding, when there is a single causal variant, the causal variant has either causal ($z \rightarrow x \rightarrow y$) or pleiotropic effect ($z \rightarrow x, z \rightarrow y$). z represents SNP, x represents gene expression and y represents phenotype. We are unable to distinguish causality from pleiotropy. When there are multiple causal variants, the effect size of x on y (b_{xy}) estimated at each causal variant can be various, linkage or multiple causal variants with distinct b_{xy} . It is a complicated scenario, which is not easy to interpret.

Responses: We fully agree with the reviewer that it is not possible to distinguish causality from pleiotropy when there is only a single variant. We also agree with the reviewer that it can be a complicated scenario when there are multiple variants, especially when one attempts to build a model through the marginal effect size estimates (b_{xy}) as used in SMR/GSMR. However, we found that it is much easier, at least conceptually, to think on modeling the causal effects through our likelihood framework in Equations 1-4. Specifically, the causal effects are now represented by the vector β ; the horizontal pleiotropic effects are represented by the vector γ ; and linkage disequilibrium is automatically accounted for in the model as we jointly model all cis-SNPs together. You can make different modeling assumptions on β and γ , and these different modeling assumptions correspond to different existing TWAS/MR approaches (Table 1). Indeed, as the reviewer #1 also pointed out (summary statement there), our likelihood framework unifies many existing MR and TWAS methods, thus facilitating the understanding and interpretation of different MR and TWAS methods for TWAS applications.

1.2.2 The proportion of multiple underlying causal variants seems to be smaller than a single causal variant. Using PMR-Egger under the assumption of polygenic model, the causal effect can be considered to be average of b_{xy} at each causal variant. The horizontal effect can be considered to be deviation from the mean effect. In the manuscript, authors noticed that cis-SNPs were in high LD, “GSMR and MR-Egger are not feasible to obtain near independent SNPs”. Using GCTA-COJO, what is the proportion of genes that have multiple independent signals?

Responses: This comment appears to contain two independent messages that the reviewer wants to convey. We answer these two messages separately below:

First, with regard to “**The proportion of multiple underlying causal variants seems to be smaller than a single causal variant**”, we apologize for not providing enough background for existing eQTL mapping studies or TWAS that lead to this incorrect statement. As far as we are aware, current eQTL studies are all of small sample sizes. The largest eQTL mapping study we are aware of is the TOPMed eQTL mapping study, which only consists of ~2,000 individuals. With small samples, there is limited statistical power to identify independent cis-SNPs associated with gene expression. As a result, existing studies can only identify one or a few independent eQTLs for the eGenes. However, being only able to identify a small number of independent cis-eQTLs does not imply that there is only a small number of causal SNPs underlying gene expression. Indeed, following your suggestion of “**using GCTA-COJO**”, we found that, the top associated independent SNPs can only explain a small proportion of cis-heritability, suggesting that a large fraction of eQTLs remains to be discovered. Specifically, we applied GCTA-COJO⁸ to analyze all 15,810 genes in GEUVADIS data and identified the top 10 independent signals for each gene regardless of their genome-wide significance. We found that the independent signals in total can only explain a medium of 16%-53% cis-heritability (the proportion of heritability explained by cis-SNPs), with the proportion increasing with the number of top independent SNPs included (i.e. 16% when only the top SNP is included; and 53% when the top 10 independent SNPs are included). The boxplot below visualizes the proportion of cis-heritability explained (y-axis) versus the number of included independent top SNPs used in the model (x-axis) across genes.

The results in GEUVADIS data is also largely consistent with our recent work in

the GENOA eQTL mapping study (n=1,032; African American samples; manuscript to be submitted soon), where we found that the primary eQTLs can only explain a small proportion of cis-SNP heritability (median = 7.9%) and that the primary eQTLs together with additional independent eQTLs (passing the genome-wide significance and obtained through GCTA-COJO) in total can only explain a small fraction of additional cis-SNP heritability (median = 13.2%). These heritability estimation results suggest that the genetic architecture underlying gene expression is highly polygenic or even omnigenic, consistent with the previously known fact that polygenic models often outperform sparse models in gene expression prediction^{4,7} and also consistent with our current results showing that polygenic models also outperform sparse models for TWAS applications. Therefore, it is important to make polygenic/omnigenic modeling assumptions on β for TWAS applications.

Second, with regard to “**Using PMR-Egger under the assumption of polygenic model, the causal effect can be considered to be average of bxy at each causal variant. The horizontal effect can be considered to be deviation from the mean effect.**”, we are unsure if this statement is correct. This statement is certainly correct when the cis-SNPs are independent. However, cis-SNPs in the real data are all highly correlated with each other due to linkage disequilibrium. The high correlation among cis-SNPs in TWAS applications makes it almost impossible to derive any simple equivalence between the marginal effect estimates and causal effect or horizontal pleiotropic effect.

1.3.1 The unrealistic assumptions may cause issues in the simulations and real data analysis. For example, Fig 2a showed PMR-Egger had larger power than the other methods only when there was “horizontal pleiotropy”. The “horizontal pleiotropy” was not explicitly described in the manuscript.

Responses: This comment appears to contain two independent messages that the reviewer wants to convey. We answer these two messages separately below:

First, with regard to “**The “horizontal pleiotropy” was not explicitly described in the manuscript**”, we apologize for not making our modeling assumption and simulation details for the horizontal pleiotropy apparent to the reviewer in the previous manuscript. Horizontal pleiotropy occurs when the SNPs affect the outcome through paths other than the exposure. Horizontal pleiotropy is modeled using parameter vector γ as shown in Equation 3 (and 4) in the Methods section, which unifies previous ways of modeling horizontal pleiotropy (Table 1). In the simulations, we simulated γ in multiple different ways to capture various possible horizontal pleiotropic effects. Following your comment, we have now highlighted the simulation section for γ in red to make it apparent to the reviewer (lines 263-270 on page 13).

Second, with regard to “**The unrealistic assumptions may cause issues in the**

simulations and real data analysis. For example, Fig 2a showed PMR-Egger had larger power than the other methods only when there was “horizontal pleiotropy””, we do not follow the logic of this statement. Fig 2a clearly shows that PMR-Egger has much higher power than SMR (and PrediXcan) across all settings, supporting the importance of making polygenic assumptions on β and modeling all cis-SNPs together for TWAS applications. Fig. 2a only shows that PMR-Egger has slightly less power as compared to TWAS/CoMM. As we explained previously in the manuscript, the slightly lower power of PMR-Egger as compared to TWAS/CoMM in the absence of horizontal pleiotropy is expected: this is not related to our polygenic modeling assumption on β (since all these three methods make a polygenic modeling assumption), but is simply due to the loss of degree of freedom inevitably occurs when we introduce parameters to model the horizontal pleiotropic term. It is well recognized in the statistical literature that any modeling assumption can lead to a small power loss when the modeling assumption is not satisfied – after all, there is no free lunch with any statistical models. Because of the additional term to model horizontal pleiotropy, PMR-Egger will lose degrees of freedom in the absence of horizontal pleiotropy. Following your comment, we have now also highlighted this explanation in red to make it apparent to the reviewer (lines 506-512 on page 24).

1.3.2 The magnitude of horizontal pleiotropy was small, 1e-4 to 2e-3, which can be supposed to be deviation from the average bxy.

Responses: As we previously stated in the Methods section, we used the horizontal pleiotropy values in the range of 1e-4 to 2e-3 as these values are the estimates obtained from real data. Using larger values does not influence the performance of our method but will make the p-value inflation of all other methods much more extreme. Following your comment, we have now highlighted the reasoning for choosing these γ in red to make it apparent to the reviewer (lines 267-270 on page 13).

(As mentioned in the responses to your earlier comment 1.2.2, we are unsure how the horizontal pleiotropy can easily be linked to marginal SNP effect estimates when SNPs are correlated with each other. Therefore, we are unsure whether the horizontal pleiotropy effect can be simply viewed as a deviation from the average marginal effect estimates bxy.)

1.3.3 In the real data analysis, many genes described in the main text were from MHC region, e.g. C2, HLA-F, ZKSCAN4. The LD is complicated in MHC region, which is usually excluded in the analysis.

Responses: Thank you and we fully agree. Following your comment, we have removed the MHC region in all our analyses in the updated manuscript to avoid

complication. Subsequently, we have provided new causal gene examples in the Results section, with detailed molecular mechanisms and previous literature support on how these identified genes may have causal effects on the phenotypes (lines 661-697 on pages 31-32).

1.3.4 SMR results showed large p-values for HERPUD1 (p=0.1), ADAM15 (p=0.14) and DNAJC27-AS1 (p=0.05), which indicates GWAS p-values at the 3 top cis-SNPs are large. More analysis might be required to investigate whether genes have causal effects on phenotypes.

Responses: Thank you and we fully agree. It is indeed hard to investigate these genes due to a lack of existing molecular biology literature on these genes. Therefore, we now provide three new gene examples in the updated Results, with detailed molecular mechanisms and previous literature support on how these identified genes may have causal effects on the phenotypes (lines 661-697 on pages 31-32).

1.3.5 In terms of MAPT, I don't think the horizontal pleiotropy can be concluded from the non-significance of PrediXcan analysis conditional on the flanking gene. Suppose two regression models, a) $y = g_1 + e$ and b) $y | g_2 = g_1 + e$, where $y | g_2$ represents conditioning y on gene 2. I don't think the non-significance of model (b) would indicate the horizontal pleiotropy. Is it because two genes are highly correlated?

Responses: Thank you and we apologize for a potential misuse of the terminology “horizontal pleiotropy” that causes your confusion. As reviewer #1 also pointed out (8th comment there), the presence of true horizontal pleiotropy is only one of the two possible explanations for these two examples where the associations go away after controlling for nearby genes. The other alternative explanation for these examples is that the SNPs used in the model are simply tagging nearby eQTLs and exhibiting their apparent “horizontal pleiotropy” through the neighboring causal genes. Therefore, in the updated text, we fully acknowledge that, in these two examples we focused on, while it is possible that SNPs display true horizontal pleiotropic effects through the neighboring gene, it is equally or more likely that SNPs used in the model are simply tagging nearby eQTLs of the causal gene and thus displaying apparent “horizontal pleiotropic effects” through the neighboring gene. Subsequently, the horizontal pleiotropic effect term in PMR-Egger may represent the apparent “horizontal pleiotropic effects” through SNP tagging to the nearby eQTLs of the causal gene, rather than the truly horizontal pleiotropic effect acted through other molecular pathways. Regardless of the interpretation of the horizontal pleiotropic effect term, however, we found it reassuring that by modeling the horizontal pleiotropic effect

term in PMR-Egger can reduce false discoveries in the case of SNP tagging. We have updated the Results section to mention this important point (lines 771-784 on page 36).

1.3.6 without any regional figures for real data analysis, I am not convinced the identified genes have causal effects on phenotypes.

Responses: Thank you for your kind reminder. We have now provided regional figures in the form of LoucsZoom plots for these real data examples (Supplementary Figures 33-35). In addition, we have now provided detailed molecular mechanisms and previous literature support on how these identified genes may have causal effects on the phenotypes in the updated Results section (lines 661-697 on pages 31-32).

2. The method requires the SNPs associated with gene expression. What is the threshold to select SNP instruments?

Responses: We apologize for the miscommunication here. Our method follows existing TWAS methods (e.g. TWAS, TIGAR, DPR, CoMM etc.) and uses all cis-SNPs. Therefore, unlike SMR, our method does not perform SNP selection and does not use any selection threshold. We added a couple sentences in the Methods section to clarify this issue (lines 149-151 on page 8; lines 399-400 on page 19).

3. Fig 1 showed p-values for SMR were strongly deflated, and p-values for LDA MR-Egger were largely inflated, which looks weird. P-value for SMR may be deflated, because SE is an approximation. Is the strong deflation because of weak instruments? The LDA MR-Egger is not too different from TWAS. For LDA MR-Egger, $b_{xy} = \text{cov}(g_x, g_y)/h^2_x$, where $\text{cov}(g_x, g_y) = b_{zx} * V^{-1} * b_{zy}$, and $h^2_x = b_{zx} * V^{-1} * b_{zx}$. For TWAS, $rg_{xy} = Z_{twas}/\sqrt{n * h^2_y} = b_{zx} * V^{-1} * b_{zy} / \sqrt{b_{zx} * V^{-1} * b_{zx} * b_{zy} * V^{-1} * b_{zy}} = \text{cov}(g_x, g_y) / \sqrt{h^2_x * h^2_y}$.

Responses: Thank you for the comments. The deflation of SMR p-values we observed in both simulations and real data is consistent with previous literature (e.g. Figure 5 in the S-PrediXcan paper⁶). The deflation of SMR p-values is likely either due to the uncertainty in selecting the instrumental SNP (i.e. the top SNP with the largest marginal association evidence may not be the causal SNP even if there is only one causal SNP for the gene) or due to the small eQTL effects across majority of genes (i.e. the top SNP with the largest marginal association evidence can only explain a small proportion of cis-heritability, even in eGenes). We have added a short explanation for SMR p-value deflation in the revised Results section (line 453-456 on page 22).

For LDA MR-Egger, different from the reviewer's impression, LDA MR-Egger is actually very different from TWAS in terms of modeling assumption. Specifically, as we previously summarized in Table 1, LDA MR-Egger makes a fixed effect size assumption on β (the p-vector of SNP effects on gene expression) while TWAS makes a BSLMM assumption on β . The fixed effect size assumption in LDA MR-Egger is unfortunately not a good assumption for TWAS applications as the number of (highly correlated) cis-SNPs is often on the same order as the number of individuals in the gene expression study. The fixed effect assumption in LDA MR-Egger, when paired with the two-stage inference procedure that ignores the estimation uncertainty in the first stage, makes LDA MR-Egger sensitive to the collinearity induced by cis-SNP correlations caused by LD. We previously provided a short explanation on the extreme LDA MR-Egger behavior in the Results. Based on the reviewer's comments, we realized that the short explanation in the previous version of manuscript was insufficient. Therefore, in the updated manuscript, we provided a more elaborated explanation in the Results section for LDA MR-Egger (lines 456-468 on page 22). Please also refer to our response to reviewer #1's 2nd comment for more detailed explanation on LDA MR-Egger.

(As a side note, the LDA MR-Egger equation provided in your comment appears to be incorrect and appears to be a TWAS equation. For LDA MR-Egger modeling assumptions, please refer to section 2.6 in the LDA MR-Egger paper as well as their source code for details.)

4. Table 1 shows PMR-Egger can deal with both individual-level and summary-level data. I am not sure whether “both” means accounting for sample overlap or simply accepting individual-level data and summary-level data as input. The summary-level data can be obtained from GWAS analysis. Therefore, all the methods are able to handle individual-level and summary-level data without accounting for sample overlap.

Responses: We apologize for not providing this background information. We previously followed existing TWAS method papers (e.g the TWAS and S-PrediXcan papers) and used the term “summary statistics” to refer to a method accepting/using summary-level data as input. In the field of TWAS, A method using summary-level data as input is often distinguished from a method that use individual-level data as input. For example, one key innovation of TWAS over PrediXcan as stated in the original TWAS paper is its use of summary-level data as input. One key innovation of S-PrediXcan over PrediXcan as stated in the recent S-PrediXcan paper is its use of summary-level data as input. Reviewer #1 also viewed the use of summary data in our method as an important contribution (10th comment there). However, following your comment, we realize that different researchers may have different opinion on the

significance of the use of summary-level data. Therefore, we have deleted that last column of Table 1 to avoid the controversy.

5. Figure 4, 5 and 6 showed lambda for PMR-EGGER was not too different from SMR or PrediXcan, but the number of genes identified by PMR-Egger was much larger than the other methods. It is confusing to me.

Responses: We apologize for not providing this background information. The number of genes identified above a genome-wide threshold is commonly used in the literature as a measure of statistical power in real data applications. The genomic control factor lambda is commonly used in the literature to measure type I error control in real data applications, as lambda captures approximately the type I error control at the level of 0.05. Power and lambda are two different statistical terms that are not necessarily correlated with each other. A similar lambda among PMR-Egger/SMR/PrediXcan would suggest similar type I error control at the nominal level of 0.05, while a higher number of genes identified by PMR-Egger over SMR/PrediXcan would suggest its higher power. Certainly, for a data with a relatively large sample size (e.g. UK Biobank) and a trait with a highly polygenic genetic architecture, then lambda may also be influenced by power of the method in addition to its type I error control. In addition, the number of significant genes may not be a perfect measure of power in certain cases and can be influenced by lambda: a method that fails to control for type I error could yield inflated p-values, leading to a high number of false discoveries. To avoid future confusion, we have added a small section in the Supplementary Text to provide this background information (lines 235-247 on page 9).

Minor comments:

6. Page 8, “Methods – PMR-Egger overview”, I think it might be better to move the first 3 paragraphs to results.

Responses: Thank you. The first three paragraphs of the method overview contain several equations, so we are a bit worried that these paragraphs might be a bit too technical for a general audience. Therefore, instead of moving them to the Results section, we followed the main idea of your suggestion and added a short sentence at the beginning of the Results section to link the readers to the method overview subsection in Methods (line 424-425 on page 21).

7. Page 9, The authors stated “our model no longer requires the exclusion restriction condition of MR, because of fitted horizontal pleiotropy in the model”. The statement is not correct. PMR-Egger follows the assumptions of MR-Egger, e.g. INSIDE. Under INSIDE assumption, the causal effect is mediated by gene

expression.

Responses: Thank you and we apologize for the confusion in the existing literature on the term “exclusion restriction condition” that leads you to believe our previous statement was incorrect. Following your suggestion, we have reworded the sentence to distinguish the general/strong exclusion restriction assumption we used in that sentence (that instruments only influence the outcome through the path of exposure) from the weak exclusion restriction assumption that people sometimes used to call the InSIDE assumption (that instruments can influence the outcome through the path of exposure and that the instrument-exposure effects and instrument-outcome effects are independent of each other). The updated sentence is in lines 178-181 on pages 9-10.

8. Page 9, the font size for the last line is not the same.

Responses: Thank you for pointing out this error, which we have corrected in the updated manuscript.

9. Page 10, the authors stated “at least one of two assumptions is not testable in practice.” I don’t think it is correct. For assumption 1), the method requires SNPs strongly associated with gene expression, at least genome-wide significant SNPs.

Responses: Thank you. Following your comment, we have reworded the whole sentence in the updated manuscript to “*Because the causal effect interpretation of α depends on MR assumptions as well as other explicit modeling assumptions, many of which are often not testable in practice, MR analysis in observational studies likely provides weaker causality evidence than randomized clinical trials.*” (lines 183-185 on page 10).

10. Page 12, “N(0, PVE_zx/556)”, “PVE” was not described.

Responses: We apologize for not making our previous definition of PVE obvious to the reviewer. We previously defined PVE_{zx} five lines above the cited place on the same page. Following your comment, we have moved the definition of PVE_{zx} to the same line of the cited place (lines 240-241 on page 12).

11. Page 24, “increasing horizontal pleiotropy (Supplementary Fig. 2e, f)”, where are supplementary figure 2 e and f?

Responses: Thank you for pointing out an error here. It should be “**Fig. 2e, f**” instead

of “**Supplementary Fig. 2e, f**”. We have corrected this error in the revised manuscript. In addition, we have carefully examined through the entire manuscript to correct for these typos.

- 1 Gusev, A. *et al.* Integrative approaches for large-scale transcriptome-wide association studies. *Nature genetics* **48**, 245-252 (2016).
- 2 Gamazon, E. R. *et al.* A gene-based association method for mapping traits using reference transcriptome data. *Nature genetics* **47**, 1091-1098 (2015).
- 3 Yang, C. *et al.* CoMM: a collaborative mixed model to dissecting genetic contributions to complex traits by leveraging regulatory information. *Bioinformatics*, doi:10.1093/bioinformatics/bty865 (2018).
- 4 Zeng, P. & Zhou, X. Non-parametric genetic prediction of complex traits with latent Dirichlet process regression models. *Nature communications* **8**, 456 (2017).
- 5 Nagpal, S. *et al.* TIGAR: An Improved Bayesian Tool for Transcriptomic Data Imputation Enhances Gene Mapping of Complex Traits. *The American Journal of Human Genetics* (2019).
- 6 Barbeira, A. N. *et al.* Exploring the phenotypic consequences of tissue specific gene expression variation inferred from GWAS summary statistics. *Nature communications* **9**, 1825 (2018).
- 7 Zeng, P., Zhou, X. & Huang, S. Prediction of gene expression with cis-SNPs using mixed models and regularization methods. *BMC genomics* **18**, 368 (2017).
- 8 Yang, J. *et al.* Conditional and joint multiple-SNP analysis of GWAS summary statistics identifies additional variants influencing complex traits. *Nature genetics* **44**, 369 (2012).

Reviewers' comments:

Reviewer #1 (Remarks to the Author):

I would like to commend the authors for performing a considerable amount of effort in their revision. The updated version has improved in clarity, and quality.

All of my concerns have been addressed and I have no further comments at this time.

Reviewer #3 (Remarks to the Author):

Thanks a lot to the authors. I am appreciated that the authors made such detailed responses. I agree that PMR-EGGER is better than the existing methods. But I have some questions from the responses. I hope the discussion may further improve the understanding of PMR-EGGER.

1. Intercept. The authors' response reminds me that β_{zx} (SNP effects on exposure) in MR-EGGER or LDA MR-EGGER are all positive orientated. There was a discussion in terms of orientation in Burgess et al. 2017 Eur. J Epidemiol (doi: 10.1007/s10654-017-0255-x). Because the orientation is arbitrary and might affect the horizontal pleiotropy and the power of detecting causality, could the authors point out that β_{zx} are orientated or not?

2. Summary stats. As many GWAS results are published every year, methods using summary stats are commonly used. PMR-EGGER is able to utilize summary stats in addition to individual-level data. In the supplementary note, Σ_1 represents LD for cis-SNPs and Σ_2 represents LD for GWAS SNPs. If neither eQTL nor GWAS datasets is available, I guess $\Sigma_1 = \Sigma_2 = \Sigma$, where Σ represents LD of reference SNPs. Could the authors clarify it in the methods and/or supplementary note?

3. Random model. My understanding of "random model" is that PMR-EGGER corrects SNP effects for allele frequency and LD, in addition to taking the SEs of estimations into account. Therefore, there would be accumulated errors for fixed model (e.g. LDA MR-EGGER). And the accumulated errors might be large when SNPs are in high LD (fig. 1a). If my understanding is correct, it is great when LDs are consistent between reference samples and discovery datasets. But would it be a caveat if the LD from the LD reference sample is not consistent with the eQTL or GWAS data (due to imputation error or other unknown reasons)?

4. Polygenicity. I agree that the cis-eQTLs may be polygenic. I think 2 causal variants might have biologically related functions if the 2 causal variants are in LD. Thus, if all the cis-SNPs are causal variants, the SNP effects might be correlated and it may appear that there are only a few causal variants, because of LD. In the simulation setting, effects at the causal variants seem to be independent. Although it does not affect the conclusion of the simulations, could the authors clarify it or improve the simulation setting?

5. The authors showed many good examples of PMR-EGGER analysis in real data. In practice, given an eQTL dataset, if phenotype (y) is a low-prevalence disease or sample size of y is small, it is great that PMR-EGGER is able to identify more causal genes, because few significant GWAS SNPs would be detected and the power to detect gene is limited. If the phenotype is a quantitative trait, there might be millions of individuals (e.g. 23 and me, UK Biobank). There are many significant GWAS SNPs. Therefore, all the other methods may identify many genes for follow-up analysis. And PMR-EGGER may identify more genes on top of that. It would be still great. But scientists may be puzzled because it might not be easy to choose one for follow-up analysis or experiment. Could the authors discuss a bit about the PMR-EGGER in practice?

Point-by-point Responses to Reviewer 3's Comments

Thanks a lot to the authors. I am appreciated that the authors made such detailed responses. I agree that PMR-EGGER is better than the existing methods. But I have some questions from the responses. I hope the discussion may further improve the understanding of PMR-EGGER.

Responses: Thank you for your positive feedback and the constructive remaining comments. Our detailed responses to your remaining comments are provided below.

1. Intercept. The authors' response reminds me that bzx (SNP effects on exposure) in MR-EGGER or LDA MR-EGGER are all positive orientated. There was a discussion in terms of orientation in Burgess et al. 2017 Eur. J Epidemiol (doi: 10.1007/s10654-017-0255-x). Because the orientation is arbitrary and might affect the horizontal pleiotropy and the power of detecting causality, could the authors point out that bzx are orientated or not?

Responses: Thank you for the comment. Your comment here is very closely related to the comment 4 previously raised by reviewer #1. The genotype orientation is indeed arbitrary. We previously used allele frequency in the GWAS data to orient genotypes. Following the comment 4 previously raised by reviewer #1, we previously examined the robustness of PMR-Egger in the presence of incorrect genotype orientation. Specifically, we performed a set of simulations where we randomly flipped the genotype encoding at a subset of SNPs prior to the inference procedure, so that the genotype coding does not match between simulations and analysis. In the previous analyses, we found that the p values from PMR-Egger for testing causal effect behaved reasonably well across a wide range of scenarios (currently lines 150-153 on page 9; Supplementary Figure 13).

Note that we have substantially shortened the current manuscript per editor's request, so the page numbers cited in previous response letter have all been updated.

Following your comment and Burgess et al. 2017¹, we have added new simulation analysis where we oriented SNP genotypes based on its effect sign on the exposure (e.g. gene expression here). Unlike the GWAS MR settings, the sample size in the gene expression study is often very small in the TWAS setting, making it challenging to accurately determine the correct sign of SNP effects on the exposure variable. Consequently, we might expect the "positive orientation" strategy in Burgess et al. 2017 to not work very well in the TWAS setting. Indeed, we found that the 'positive orientation' approach behaves quite similarly as the allele frequency orientation approach in terms of type I error control for testing the causal effect. In addition, such 'positive orientation' approach loses power substantially compared to the allele

frequency orientation approach, presumably because the ‘positive orientation’ strategy violated our normality assumption on the SNP effects on the gene expression. These new results are shown in Supplementary Figure 14, with details provided in Results section (lines 153-155 on page 9; lines 181-183 on page 10). We also explained that we primarily used genotype coding/orientation approach in the Methods (lines 428-430 on pages 22-23) and described the positive orientation approach in the Methods (lines 551-553 on page 28).

2. Summary stats. As many GWAS results are published every year, methods using summary stats are commonly used. PMR-EGGER is able to utilize summary stats in addition to individual-level data. In the supplementary note, Σ_1 represents LD for cis-SNPs and Σ_2 represents LD for GWAS SNPs. If neither eQTL nor GWAS datasets is available, I guess $\Sigma_1 = \Sigma_2 = \Sigma$, where Σ represents LD of reference SNPs. Could the authors clarify it in the methods and/or supplementary note?

Response: Thank you for your comment. Indeed, if neither eQTL nor GWAS data is available, then $\Sigma_1 = \Sigma_2 = \Sigma$. We have added one sentence to clarify this in the supplementary note (lines 147-148 on page 6 in Supplementary Note).

3. Random model. My understanding of “random model” is that PMR-EGGER corrects SNP effects for allele frequency and LD, in addition to taking the SEs of estimations into account. Therefore, there would be accumulated errors for fixed model (e.g. LDA MR-EGGER). And the accumulated errors might be large when SNPs are in high LD (fig. 1a). If my understanding is correct, it is great when LDs are consistent between reference samples and discovery datasets. But would it be a caveat if the LD from the LD reference sample is not consistent with the eQTL or GWAS data (due to imputation error or other unknown reasons)?

Responses: Thank you for the comment. Your comment here is very closely related to the comment 10 previously raised by reviewer #1. Indeed, for any methods using summary statistics, it is important to match the LD pattern in the reference sample with that in the GWAS (or eQTL) data²⁻⁷. Previously we followed reviewer #1’s comment 10 and examined whether the inference results are sensitive to the choice of the reference panel. Specifically, we previously constructed the SNP by SNP correlation matrix from three different reference panels: all individuals from the GWAS data; 10% randomly selected individuals from the GWAS data; individuals of European ancestry from the 1,000 Genomes project. We then applied the summary statistics based approach of PMR-Egger to each reference panel and compared results with the individual level data based approach of PMR-Egger that was applied to the complete

data. Both p-values for testing the causal effects and the p-values for testing the pleiotropy effects are consistent between the summary statistics based approach and the individual data based approach. These results were previously shown in the Results section (currently Supplementary Figure 45).

Following your suggestion, we conducted additional simulations by examining the most extreme case, where we constructed the reference panel using the individuals of African ancestry from the 1,000 Genomes phase 3 project. As expected, the results obtained using summary version of PMR-Egger with the African reference panel are less consistent with the results obtained using European individual-data, at least when compared to the previous results using summary version of PMR-Egger based on the other three reference panels. The new results based on a reference panel consists of individuals of African ancestry are provided in Supplementary Figure 45, with details described in the Discussion section (lines 389-393 on page 20) and Methods section (lines 567-572 on page 29).

4. Polygenicity. I agree that the cis-eQTLs may be polygenic. I think 2 causal variants might have biologically related functions if the 2 causal variants are in LD. Thus, if all the cis-SNPs are causal variants, the SNP effects might be correlated and it may appear that there are only a few causal variants, because of LD. In the simulation setting, effects at the causal variants seem to be independent. Although it does not affect the conclusion of the simulations, could the authors clarify it or improve the simulation setting?

Responses: Thank you for the comment. Following your suggestion, we have conducted additional simulations with correlated causal effects on the gene expression. The new results are largely consistent with the main results where causal effects are independent of each other. The new results are shown in Supplementary Figure 5, with details provided in the Results section (lines 134-135 on page 8; lines 200-201 on page 11) and Methods section (lines 537-539 on page 27).

5. The authors showed many good examples of PMR-EGGER analysis in real data. In practice, given an eQTL dataset, if phenotype (y) is a low-prevalence disease or sample size of y is small, it is great that PMR-EGGER is able to identify more causal genes, because few significant GWAS SNPs would be detected and the power to detect gene is limited. If the phenotype is a quantitative trait, there might be millions of individuals (e.g. 23 and me, UK Biobank). There are many significant GWAS SNPs. Therefore, all the other methods may identify many genes for follow-up analysis. And PMR-EGGER may identify more genes on top of that. It would be still great. But scientists may be puzzled because it might not be easy to choose one for follow-up analysis or experiment. Could the authors discuss a bit

about the PMR-EGGER in practice?

Responses: Thank you for the comment. Indeed, this is a general question that we as the GWAS community should all think hard on. As you pointed out, given that there are many significant GWAS SNPs discovered at biobank scale data, which set of SNPs should we follow up experimentally? TWAS, including our study, attempts to prioritize genes for follow up studies. Increasing the power of TWAS, liked our method does, would effectively increase the number of true positives and reduce the number of false positives one would get in the top set of genes. Therefore, even we can only follow up with the top genes immediately due to resource limitation, high TWAS power can ensure that these genes being followed up are likely true positives, leading to better results replication and success of experimental validation. Finally, TWAS fine mapping methods, such as recently developed FOCUS⁸ (cited and explored in the present manuscript), can be a useful option for refining results and identify the most likely causal genes for follow up experiments. Following your comment, we have added these discussions to the Discussion section (line 340-343 on page 18). (Note again that we can only briefly discuss this issue in the text due to the strict word limitation required by the editor for this revision.)

References

- 1 Burgess, S. & Thompson, S. G. Interpreting findings from Mendelian randomization using the MR-Egger method. *Eur J Epidemiol* **32**, 377-389, doi:10.1007/s10654-017-0255-x (2017).
- 2 Yang, J. *et al.* Conditional and joint multiple-SNP analysis of GWAS summary statistics identifies additional variants influencing complex traits. *Nat Genet* **44**, 369-375, S361-363, doi:10.1038/ng.2213 (2012).
- 3 Pasaniuc, B. & Price, A. L. Dissecting the genetics of complex traits using summary association statistics. *Nat Rev Genet* **18**, 117-127, doi:10.1038/nrg.2016.142 (2017).
- 4 Bulik-Sullivan, B. K. *et al.* LD Score regression distinguishes confounding from polygenicity in genome-wide association studies. *Nat Genet* **47**, 291-295, doi:10.1038/ng.3211 (2015).
- 5 Barbeira, A. N. *et al.* Exploring the phenotypic consequences of tissue specific gene expression variation inferred from GWAS summary statistics. *Nat Commun* **9**, 1825, doi:10.1038/s41467-018-03621-1 (2018).
- 6 Duncan, L. *et al.* Analysis of polygenic risk score usage and performance in diverse human populations. *Nat Commun* **10**, 3328, doi:10.1038/s41467-019-11112-0 (2019).
- 7 Deng, Y. & Pan, W. Improved Use of Small Reference Panels for Conditional and Joint Analysis with GWAS Summary Statistics. *Genetics* **209**, 401-408, doi:10.1534/genetics.118.300813 (2018).
- 8 Mancuso, N. *et al.* Probabilistic fine-mapping of transcriptome-wide association studies. *Nat Genet* **51**, 675-682, doi:10.1038/s41588-019-0367-1 (2019).

REVIEWERS' COMMENTS:

Reviewer #3 (Remarks to the Author):

Thank you very much for the authors' response. It is very clear to me and I do not have any further question.